# Synthetic modeling reveals HOXB genes are critical for the initiation and maintenance of human leukemia

Manabu Kusakabe [1,5], Ann Chong Sun[1,6], Kateryna Tyshchenko[1,6], Rachel Wong[1], Aastha Nanda[1], Claire Shanna[1], Samuel Gusscott[1], Elizabeth A. Chavez[2], Alireza Lorzadeh[3], Alice Zhu[3], Ainsleigh Hill[1], Stacy Hung[2], Scott Brown[4], Artem Babaian[1], Xuehai Wang[1], Robert A. Holt[4], Christian Steidl[2], Aly Karsan[4], R. Keith Humphries[1], Connie J. Eaves [1], Martin Hirst [3,4] & Andrew P. Weng[1]

Mechanistic studies in human cancer have relied heavily on cell lines and mouse models, but are limited by in vitro adaptation and species context issues, respectively. More recent efforts have utilized patient-derived xenografts; however, these are hampered by variable genetic background, inability to study early events, and practical issues with availability/reproducibility. We report here an efficient, reproducible model of T-cell leukemia in which lentiviral transduction of normal human cord blood yields aggressive leukemia that appears indistinguishable from natural disease. We utilize this synthetic model to uncover a role for oncogene-induced HOXB activation which is operative in leukemia cells-of-origin and persists in established tumors where it defines a novel subset of patients distinct from other known genetic subtypes and with poor clinical outcome. We show further that anterior HOXB genes are specifically activated in human T-ALL by an epigenetic mechanism and confer growth advantage in both pre-leukemia cells and established clones.

[1] Terry Fox Laboratory, BC Cancer Agency, Vancouver V5Z 1L3 BC, Canada. [2] Lymphoid Cancer Research, BC Cancer Agency, Vancouver V5Z 1L3 BC, Canada. [3] Department of Microbiology and Immunology, Michael Smith Laboratories, University of British Columbia, Vancouver V6T 1Z4 BC, Canada. [4] Canada's Michael Smith Genome Sciences Centre, BC Cancer Agency, Vancouver V5Z 1L3 BC, Canada. [5] Present address: Department of Hematology, Faculty of Medicine, University of Tsukuba, Tsukuba 305-8575, Japan. [6] These authors contributed equally: Ann Chong Sun, Kateryna Tyshchenko. Correspondence and requests for materials should be addressed to A.P.W. (email: aweng@bccrc.ca)

Despite the many and important conceptual insights mouse models have brought to our understanding of human cancer, they are, by definition, incapable of revealing mechanisms that are dependent on human-specific elements. It is clear there are important differences between mouse and human cells, particularly with respect to cellular transformation[1]. Particularly noteworthy is the observation that 40–90% of transcription factor binding events are not conserved between mouse and human[2,3]. Importantly, the translational impact of findings derived from cancer models based on transformation of mouse cells is immediately hampered by the need to verify if and to what extent the identified molecular mechanisms remain similarly operative in human cells. In the case of the hematopoietic system, the complement of cell surface markers used to define hematopoietic stem cells in mouse and human are completely different[4], thus limiting for instance the ability to translate work on leukemia stem cells identified in mouse models to human disease.

To mitigate species-specific limitations, many studies now incorporate validation of findings using patient-derived xenografts (PDX). Short of in-patient clinical trials, PDX models currently represent the closest we can get to bona fide human disease in terms of a platform for functional studies[5,6]. Of course, established human cell lines have and will continue to provide valuable insights into molecular mechanisms, but suffer the well-recognized caveat of rigorous selection for growth in vitro that can distort, and thus may not be representative of natural biological processes. The extent of genetic variation present in large PDX collections, however, both in terms of the mutational complement in each tumor and the genetic background of each patient, raises daunting challenges to understanding the mechanistic contribution of individual genetic elements and how they manifest on varied genetic/mutational backgrounds. Finally, neither established cell lines nor PDX models are able to functionally interrogate the earliest of molecular events as oncogenes redirect cells from normal to malignant developmental trajectories.

We thus sought here to take a synthetic approach and create custom-designed tumors using prespecified combinations of genetic elements. We opted for normal human cells as starting material in order to study the process of malignant transformation from beginning to end, and used multipotent hematopoietic progenitor/stem cells from umbilical cord blood (CB) as they are a consistent and renewable resource. We attempted to create synthetic T-cell acute lymphoblastic leukemia (T-ALL) as the genetics have been well described by landscape sequencing[7,8] with several oncogenes and tumor suppressors validated by transgenic mouse studies[9]. The major genetic classes of T-ALL involve TLX1/3, TAL1/SCL, LMO1/2, LYL1, CALM-AF10, SET-NUP214, and NOTCH1 as defined by chromosomal translocation or over/contextually inappropriate expression[10]. Importantly, tumor suppressors p16INK4a and p14ARF are deleted/silenced in over 80% of cases[11] and thus represents a near-requisite event for T-ALL establishment. We delivered specified combinations of these various oncogenes into CD34+ CB cells by lentiviral transduction, followed by culture in vitro on OP9-DL1 feeders to examine molecular events occurring in the initial stages of malignant reprogramming of normal T-cell progenitors, and then injection into immunodeficient mice to score for leukemogenesis in vivo.

## Results

### Transduced oncogenes drive expansion of CB cells in vitro.
We sought here to create human T-ALL de novo from normal CD34+ CB progenitors by lentiviral transduction with a combination of known T-ALL oncogenes. We combined activated NOTCH1 (NOTCH1ΔE) with LMO2/TAL1, LYL1, TLX1, TLX3, HOXA9, MEF2C, and NKX2.1, which were marked with GFP and Cherry

fluorescent reporters, respectively (Fig. 1a). We included BMI1 with each of the Cherry vectors on the premise its transcriptional repression/silencing of CDKN2A, which encodes both p16INK4a and p14ARF[12], would be critical for T-ALL establishment. BMI1 has also been identified as essential for self-renewal of hematopoietic, neural, and intestinal stem cells[13]. Coexpression of multiple genes from a single lentivirus was accomplished by linking cDNAs with picornaviral 2A sequences[14]. Transduced cells were passaged on OP9-DL1 stromal feeders every 4–5 days to study their behavior in vitro. DL1-expressing feeders were utilized so that nontransduced control cells would undergo early T-cell differentiation[15] and thus serve as a close comparator for effects of the delivered oncogenes. Importantly, cells cultured in this manner maintain the ability to engraft live animals and contribute to immune reconstitution[15]. Strikingly, doubly transduced GFP+ Cherry+ (hereafter referred to as G+C+) cells progressively outcompeted singly- and nontransduced populations in vitro for six of seven assayed gene combinations, comprising the majority of cells within 30–50 days (Fig. 1b, c). Of note, G+C− cells expanded for the first few weeks, but were outcompeted thereafter by G+C+ cells (Fig. 1c). Further, in cultures transduced with NOTCH1-only, G+ cells expanded initially, but regressed somewhat and never grew to exceed G− cells (Supplementary Fig. 1). In the NOTCH1ΔE (N) + LMO2/TAL1/BMI1 (LTB) gene combination, G+C+ cells typically attained ~$10^6$-fold expansion by day 50 (Fig. 1d) and exhibited an immature CD34+/− CD38+ CD7+ CD1a− CD2− sCD3− T-cell phenotype whereas nontransduced cells in the same cultures differentiated further to a CD34− CD1a+ stage (Fig. 1e, Supplementary Fig. 2)[16].

### Transduced CB cells produce lethal T-cell leukemias in vivo.
To score for leukemia-initiating activity in vivo, transduced CB cells cultured up to 25 days in vitro on OP9-DL1 feeders were injected into NSG mice. In initial protocols, human CD45+ cells were FACS sorted from day 10 cultures and injected intrahepatically into sublethally irradiated neonatal recipients[17]. Of note, the injected hCD45+ cells included a mixture of nontransduced (G−C−), singly transduced (G+C− and G− C+), and doubly transduced (G+C+) populations (Fig. 1c). Subsequent protocols involved sorting of doubly transduced CB cells (hCD45+ G+C+) from day 24–25 cultures and intravenous injection into adult recipients. As our data are most mature for the N+ LTB gene combination, we will focus here on those results.

We obtained malignant leukemias with T-ALL-like features in 36/43 primary recipients from seven different N+ LTB transduction experiments with overall median latency of 161 days (range 79–321 days) (Fig. 2a, Supplementary Data 1). Clinically morbid animals typically exhibited hepatosplenomegaly, lymph node and thymic masses, hypercellular bone marrow with extensive infiltration by leukemic blasts, and circulating leukemia cells with immature blast-like cytomorphology (Fig. 2b). Tumors also exhibited clonal TCRG rearrangements as assessed by clinical BIOMED-2 assay[18] (Fig. 2c).

In 22/24 recipients injected with hCD45+ G+C+ cells (FACS sorted from day 24–25 N+ LTB-transduced CB cultures), we obtained G+C+ leukemias of T-cell lineage, typically CD7+ CD2+ sCD3+/− CD1a+/− and variable CD4/CD8 pattern including CD4− CD8− (DN), CD4+ CD8+ (DP), and CD4− CD8dim (SP8dim) (Supplementary Fig. 3, Supplementary Data 1). Among 19 recipients of hCD45+ cells (FACS sorted from day 10–11 N+ LTB-transduced CB cultures), seven mice developed G+C+ leukemias, seven developed G+C− leukemias, and one showed persistent low-level G+C+ engraftment (1–2% in PB) (Supplementary Data 1). Whereas G+C+ leukemias demonstrated a spectrum of CD4/CD8 phenotypes, G+C− leukemias were

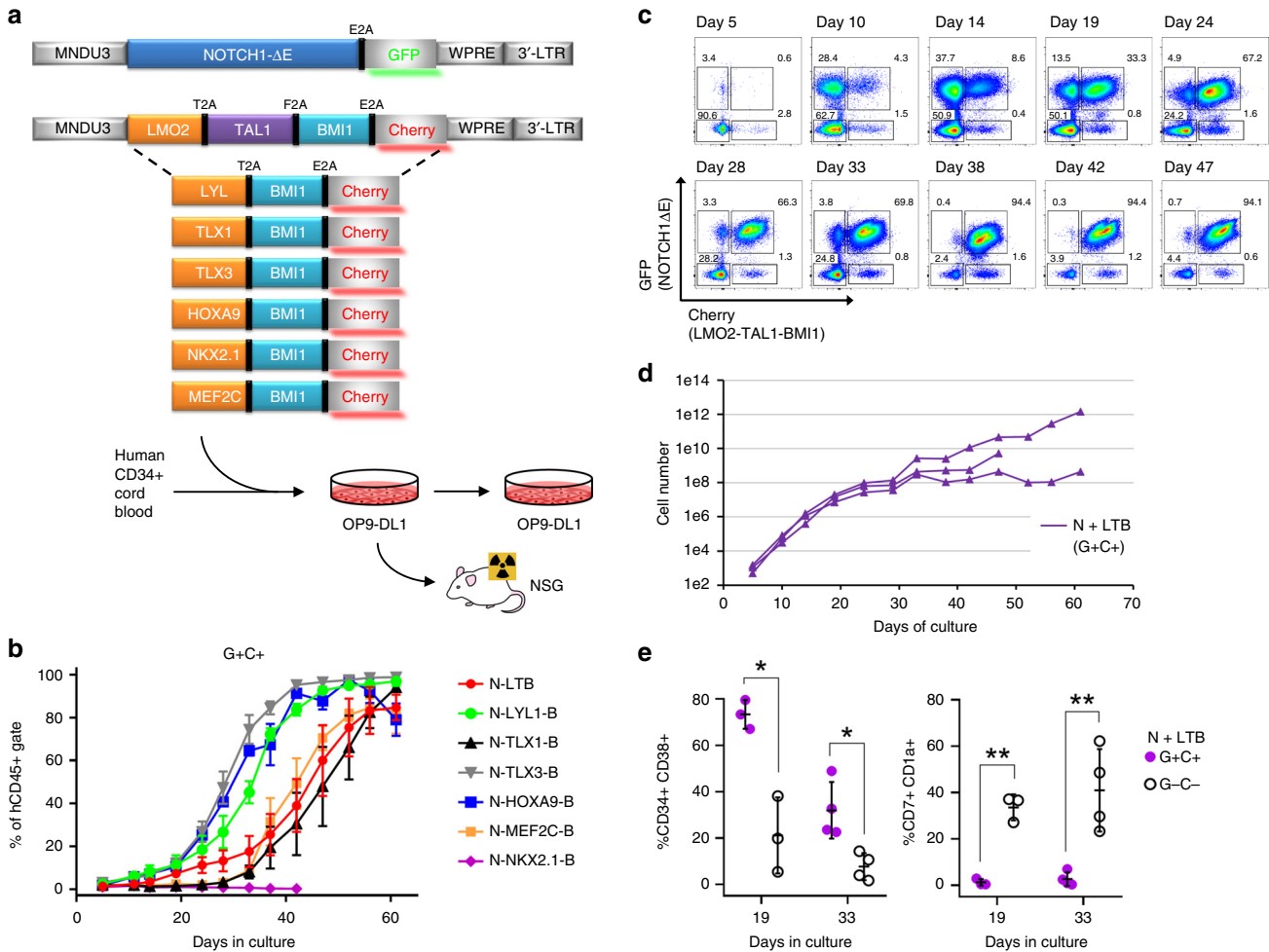

**Fig. 1** NOTCH1 combined with other T-ALL oncogenes drives expansion of CB cells in vitro. **a** Experimental schema. Human CD34+ cord blood cells were transduced with GFP and Cherry lentiviral constructs, then plated on OP9-DL1 stromal feeders for varying periods of time prior to transplantation into immunodeficient NSG mice. **b** Flow cytometric tracking of doubly transduced (GFP+Cherry+, or G+C+) CB cells at each serial passage onto fresh OP9-DL1 feeders every 4–5 days. CB cells were transduced with lentiviral particles on days 0 and 5 only. Mean values ± range for duplicate experiments are plotted. **c** Flow cytometric tracking of NOTCH1ΔE-GFP, LTB-Cherry-transduced CB cells cultured as in (**b**). Gated live, hCD45+ events are depicted. Data are representative of multiple replicates. **d** Absolute growth of NOTCH1ΔE-GFP, LTB-Cherry doubly transduced cells (G+C+). Total cell number is extrapolated based on fold-increase in cell yield at each passage. Growth curves from three independent experimental replicates are shown. **e** Flow cytometric immunophenotyping of NOTCH1ΔE-GFP, LTB-Cherry doubly (G+C+) and nontransduced (G−C−) CB cell populations from days 19 and 33. Each plotted datapoint represents a separate culture. Mean ± SD values are plotted. *$p < 0.05$; **$p < 0.01$ (two-tailed $t$ test with Holm−Sidak correction for multiple comparisons)

mostly DP, but did include one DN case (6 DP + 1 DN in total; Supplementary Fig. 3, Supplementary Data 1). Given the ~6-fold greater abundance of G+C− over G+C+ cells in day 10–11 hCD45+ inocula (Fig. 1c, Supplementary Data 1), these findings suggest there is selection for the full NLTB (G+C+) oncogenic payload in vivo similar to that observed in vitro, although N alone (G+C−) is also capable of producing aggressive leukemia in primary recipients.

**CB leukemias with both N and LTB are readily transplantable.** We next tested primary leukemias for their ability to produce disease in secondary recipients. We tested five different primary G+C+ leukemias and one low-level engrafted, but nonlethal case, and found all six to produce lethal G+C+ leukemias in secondary recipients (Fig. 2d, Supplementary Data 2). We also tested four different primary G+C− leukemias and sorted G+C− cells from a primary recipient harboring both G+C+ and G+C− subpopulations, and found only two of the five to produce lethal G+C− leukemias in secondary recipients (Fig. 2e, Supplementary

Data 2). Thus, whereas the combination of N + LTB consistently yielded fully transformed, serially transplantable leukemias, N alone was less efficient in doing so, yielding lethal, yet non-self-renewing lymphoid expansions in about half of instances (Fig. 2e, 3/5 clones). We also performed limiting dilution transplants to compare leukemia-initiating cell (LIC) frequencies in transplantable G+C+ vs. non-transplantable G+C− leukemias. Whereas the G+C+ primary leukemia from mouse #62 exhibited an LIC frequency of greater than 1 in 4100 cells, the G+C− primary leukemia from mouse #63 showed an LIC frequency of less than 1 in 1,200,000 cells, for a difference of at least ~300-fold (Supplementary Fig. 4). These data reveal that while both N + LTB and N-only can produce serially transplantable (i.e., self renewing) leukemias, this process occurs with much lower efficiency when only N is provided, presumably due to the need for stochastic acquisition of additional oncogenic hits.

**Transduced CB cells show no evidence of clonality in vitro.** To ascertain what stage of leukemogenic transformation the

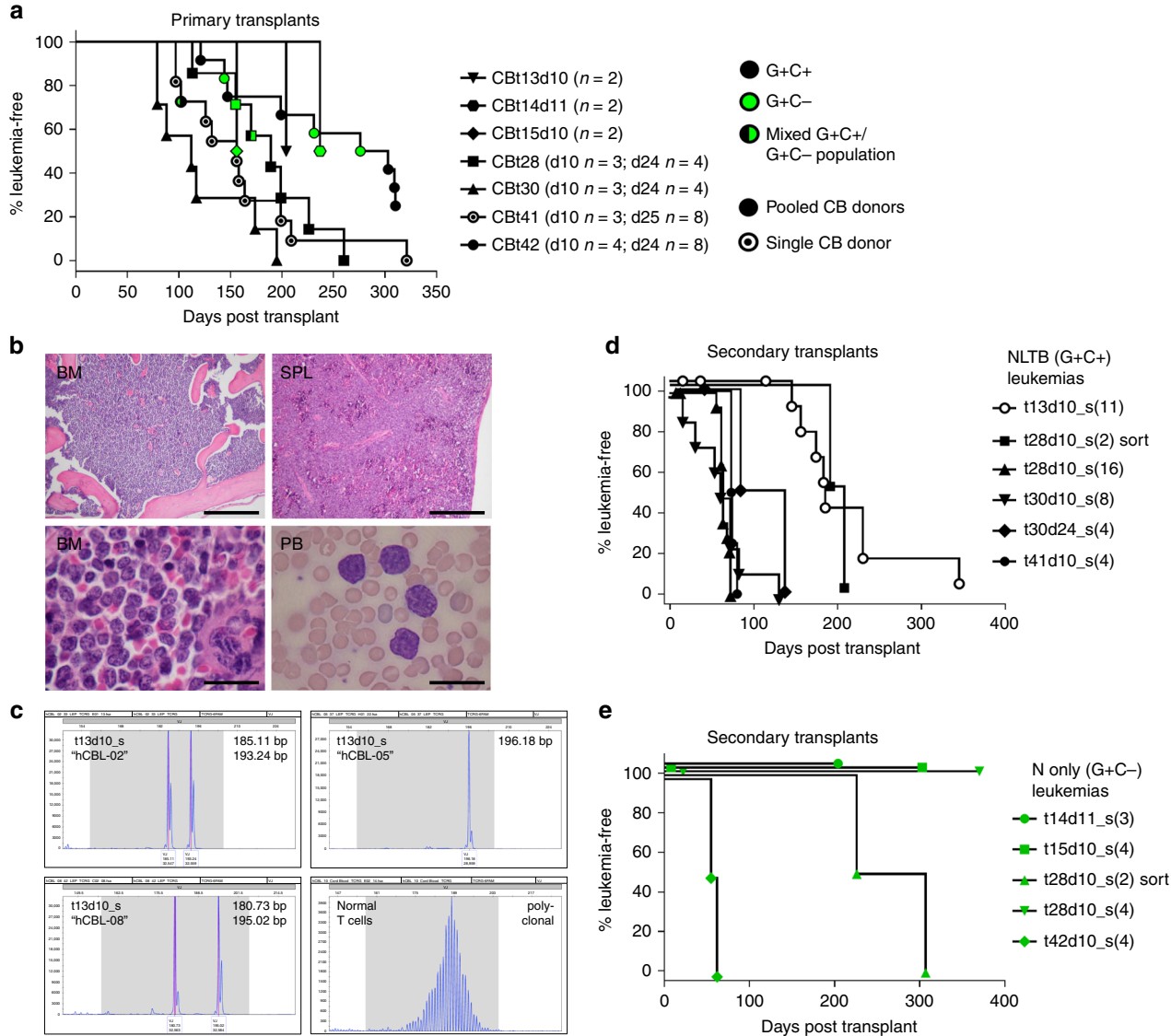

**Fig. 2** De novo transformation of CB cells by NOTCH1 plus LMO2-TAL1-BMI1. **a** Kaplan−Meier survival curves for primary recipient mice. Mice were injected with CB cells transduced with N(GFP) + LTB(Cherry) lentiviruses. Data from seven independent experimental trials are depicted with $n$ recipient mice per trial. All leukemic animals with the exception of trial 13 (CBt13) achieved clinically morbid disease endpoints requiring euthanasia. G GFP, C Cherry. **b** Formalin-fixed, paraffin-embedded (FFPE) tissue histology and air-dried peripheral blood smear morphology of NLTB CB leukemias. Representative fields of tissues from multiple G+C+ leukemic animals are shown. Scale bar = 1 mm (BM upper), 20 μm (BM lower), 0.5 mm (SPL), 20 μm (PB). BM bone marrow, SPL spleen, PB peripheral blood. **c** BIOMED-2 TCRG clonality assay. G+C+ cells from spleens of mice with NLTB CB leukemias were FACS sorted and genomic DNA extracted for analysis. Dominant peaks indicative of clonal T-cell populations are identified in each of three NLTB CB leukemias (CBL) shown. The normal T-cell control shows a Gaussian distribution of peak sizes indicative of a polyclonal T-cell population. **d**, **e** Kaplan −Meier survival curves for secondary recipient mice transplanted with primary **d** NLTB CB leukemias (G+C+) or **e** N-only CB leukemias (G+C−). Six and five different primary leukemias, respectively, were assayed. Number of recipients for each transplantation experiment are shown in parentheses. All leukemic animals achieved clinically morbid disease endpoints requiring euthanasia

transduced CB cells had attained in vitro prior to transplantation, we applied the clinical BIOMED-2 assay to detect and track clonal TCRG rearrangements. TCRG profiles of in vitro-expanded cells showed no convincing evidence of a dominant, clonally rearranged population in any of the four different CB trials tested (Fig. 3a). Further, comparison of TCRG profiles from in vitro-expanded CB cells and the leukemias they produced after transplantation revealed no clear evidence for perdurance of a dominant clone (Fig. 3b, Supplementary Fig. 5). As well, leukemias arising in animals transplanted with the same pool of transduced CB cells exhibited distinct clonal TCRG rearrangements (Figs. 2c, 3b; Supplementary Fig. 5) and also distinct donor

STR patterns (Fig. 3c, Supplementary Fig. 6), suggesting that dominant, clonally rearranged leukemias had not already arisen in vitro prior to transplantation.

We also submitted gDNA for commercial ImmunoSEQ TCRG assay from transduced CB cells that had been expanded in vitro for 14–38 days. Analysis of CDR3 fragment lengths revealed normal Gaussian distributions for both transduced G+C+ cells and G−C− controls (Supplementary Fig. 7). Read counts were about 5–6× higher for G−C− samples, consistent with a higher proportion of cells progressing through early T-cell development. Tracking of individual rearrangements showed no evidence of dominant clones emerging over time in culture among G+C+

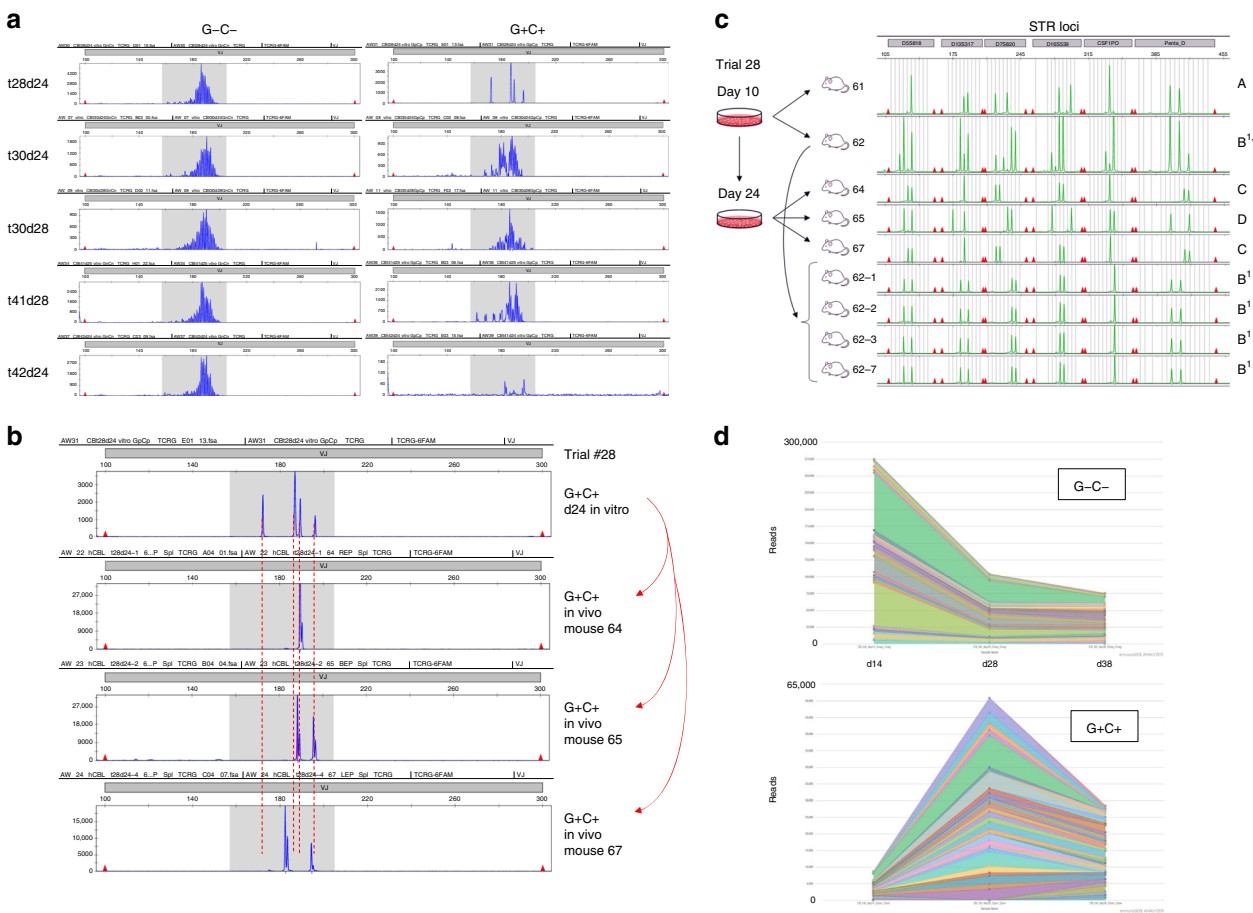

**Fig. 3** NLTB-transduced CB cells do not develop into clonally rearranged leukemias in vitro. **a**, **b** BIOMED-2 TCRG clonality assay. **a** CB cells were transduced with N + LTB lentiviruses and cultured on OP9-DL1 feeders for up to 28 days. Doubly transduced (G+C+) and nontransduced (G−C−) subsets of cells were FACS sorted and genomic DNA extracted for analysis. No dominant peaks indicative of clonal T-cell populations were identified in any of these samples. **b** Doubly transduced (G+C+) CB cells were FACS sorted from day 24 in vitro cultures, as well as from the leukemic mice into which the cultured cells were injected. Genomic DNA was then extracted, and the distribution of amplified TCRG DNA fragments analyzed by GeneScan (Applied Biosystems). Dotted red lines are overlaid to facilitate comparison of peak sizes between samples. Peak sizes are reproducible within less than 0.5 bp. **c** STR profiling of CB leukemias. Genomic DNAs from primary CB leukemias were profiled by Promega PowerPlex 16HS assay. STR patterns from four different individual donors (A, B1, C, D) can be discerned. The minor B2 pattern was not represented after serial transplant. **d** ImmunoSEQ TCRG clonality assay. CB cells were transduced with N + LTB lentiviruses and cultured on OP9-DL1 feeders. Culture samples were taken at 14, 28, and 38 days. Doubly transduced (G+C+) and nontransduced (G−C−) subsets were FACS sorted and genomic DNA extracted for ImmunoSEQ TCRG (Survey) analysis by Adaptive Biotechnologies. Stacked reads are plotted for the 25 most frequent rearrangements with each alternating color indicating a unique, clonally rearranged CDR3 DNA fragment

cells, with the clone distribution pattern appearing highly similar to that of control G−C− cells (Fig. 3d). Taken together, these results support the interpretation that NLTB-transduced CB cells do not progress to the point of dominant clonal populations for up to 38 days in culture, and thus do not represent clonal leukemias prior to transplantation.

**CB leukemia can be generated from multiple different donors**. For most of these experiments we used pooled CB cells from hundreds of donors. To determine from how many donors we were able to generate leukemias, we performed STR profiling on genomic DNA isolated individual leukemic mice. From two independent experimental trials that used cells from same CB pool, we were able to identify leukemias with at least five different STR profiles (four donors A−D shown in Fig. 3c, fifth donor F shown in Supplementary Fig. 6). Six of seven experimental trials used cells from three different CB pools, while the seventh was performed using CB cells from a single donor (Supplementary

Data 1). In total, we were able to generate leukemias from at least eight different individual CB donors, supporting that the ability of N + LTB to transform CB cells is not limited to rare individuals in the population.

**CB leukemias show clonal evolution including NRAS mutation**. We performed whole exome sequencing on a set of three lethal G+C+ leukemias from secondary recipient mice which had all been injected with cells from a primary recipient that showed persistent low-level engraftment by G+C+ cells, but was otherwise healthy. Analysis of single nucleotide variants (SNVs) confirmed the three leukemias had derived from a common ancestral clone as evidenced by 542 common SNVs. Interestingly, 2/3 were highly similar to one another, but the third showed substantial divergence with 3063 private SNVs (Fig. 4a). One of the private SNVs in this third leukemia was an NRAS gain-of-function mutation, which occurs commonly in human T-ALL[7,8,19], and thus these findings would suggest that clonal

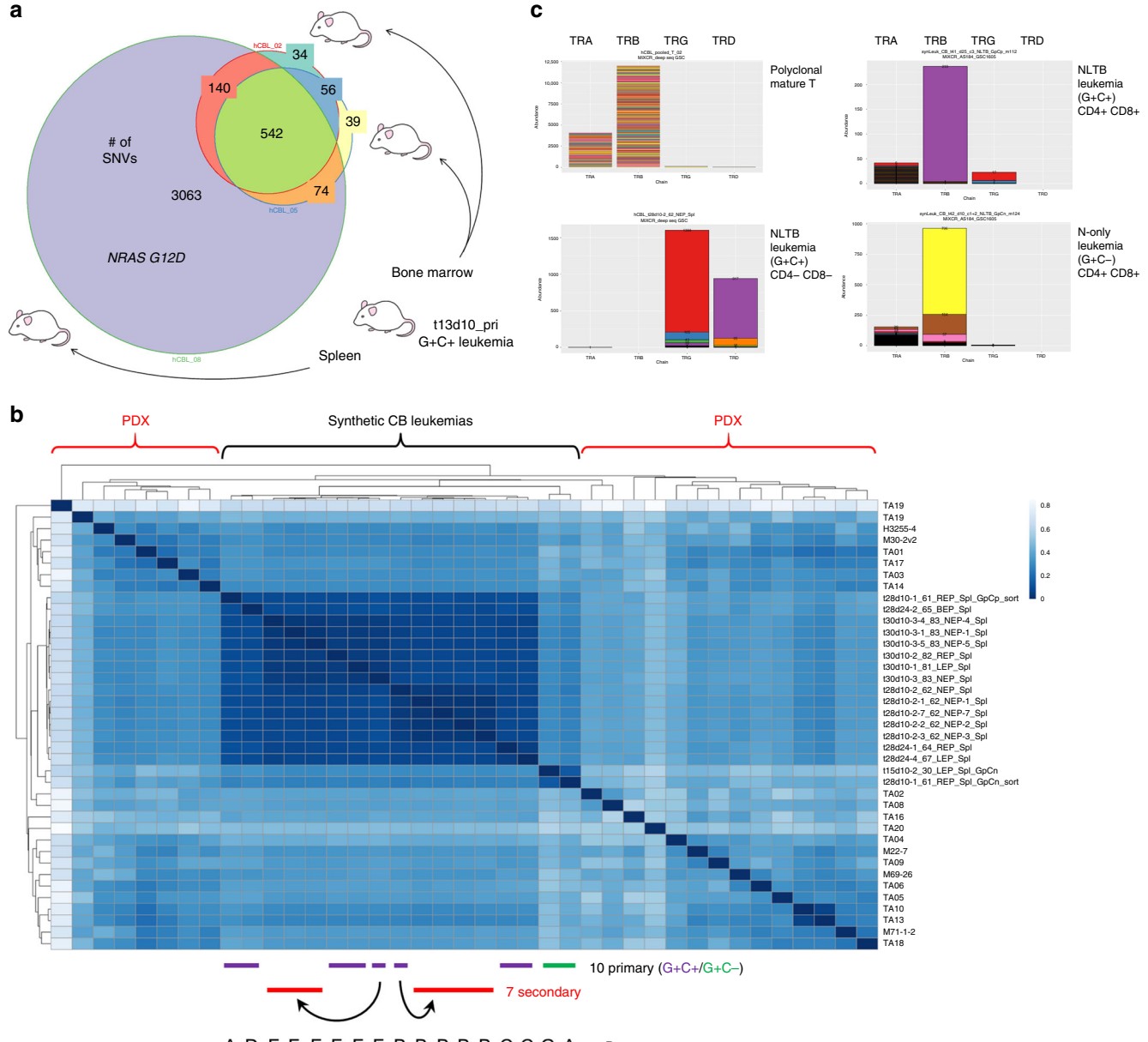

**Fig. 4** Synthetic CB leukemias exhibit clonal evolution in vivo and are highly similar to PDXs. **a** Venn diagram of single nucleotide variants (SNVs). FACS-sorted G+C+ cells from bone marrow or spleen of a low-level engrafted primary recipient mouse were injected into secondary recipients. G+C+ cells were FACS sorted from clinically morbid secondary recipients and genomic DNA extracted for whole exome sequencing. **b** Unsupervised hierarchical clustering of synthetic NLTB CB leukemias and PDX T-ALLs based on RNA-seq data. Following batch correction with ComBat, correlation distances (1 − Spearman coefficient) across 39 samples (17 synthetic CB leukemias + 22 PDX) were calculated using the top 1000 variable genes within the PDX sample set. Individual CB donors were discriminated by STR profiling. Darker blue indicates greater similarity. PDX RNA-seq data are from the PRoXe repository (NCBI SRA Accession SRP103099). Color scale indicates (1 − Spearman) correlation distance. Darker blue indicates greater similarity. **c** TCR clonality by MiXCR analysis of RNA-seq data. Distinct CDR3 regions are indicated by alternating colors within each sample; matching colors across samples are coincidental and do not indicate sequence identity. Mature T-cell control shows polyclonal TRA/TRB. The depicted G+C+ CB leukemias show mono/oligoclonal patterns for TRG/TRD and TRB, respectively. The depicted G+C− CB leukemia shows oligoclonal TRB

evolution similar to that which occurs in natural disease is operative in this synthetic model.

**RNA-seq reveals CB leukemias are highly similar to PDXs.** To determine how similar/dissimilar the synthetic leukemias were compared to bona fide human T-ALL, we performed RNA-seq on a set of 17 CB leukemias and compared them to a collection of 22 different PDX T-ALLs[6]. Using the top 1000 variable genes among PDX samples and unsupervised hierarchical clustering (Fig. 4b),

we observed that G+C+ leukemias were highly similar to one another, even across different experimental trials and originating from different donors, and that G+C− leukemias clustered separately, indicating that inclusion of LTB alters the transcriptional signature. Importantly, correlation distances between CB leukemias and individual PDXs fell within the range of distances between individual PDXs, revealing that CB leukemias reside within the spectrum of natural T-ALL variation and do not appear instead as distant outliers. We would interpret these data to support that synthetic leukemias are highly reproducible and

represent a reasonable approximation of naturally occurring human T-ALL.

We also performed unsupervised hierarchical clustering using the top variable genes among just NLTB G+C+ leukemias to determine how they differed from one another (Supplementary Fig. 8). The most notable feature was that leukemias clustered according to donor as defined by STR profiling. Since each donor leukemia represents by definition a distinct cell-transduction event, we would conclude that the source of greatest variation among synthetic NLTB leukemias is genetic/epigenetic background of the donor and/or viral integration effects.

We also extracted CDR3/V(D)J junctional reads for each of the TCR loci from RNA-seq data using the MiXCR software package[20]. We found CB leukemias to show mostly mono/oligoclonal TCR rearrangements with G+C+ leukemias tending to express rearranged TCRG and TCRD when of CD4− CD8− (DN) phenotype, and rearranged TCRB when of CD4+ CD8+ (DP) phenotype. G+C− leukemias, which were typically DP phenotype, tended to express rearranged TCRB (Fig. 4c). Together with immunophenotyping data, these findings suggest that while NLTB (G+C+) leukemias likely span both pre- and post-β-selection stages of T-cell development, N-only (G+C−) leukemias are most often post-β-selection[21].

**HOXB genes are upregulated in nascently transduced CB cells.** The in vitro component of the CB model allows direct access to transduced cells as they undergo the very first molecular changes as they are redirected from normal to malignant developmental trajectories. Accordingly, we harvested cultures at various time points and FACS-sorted singly and doubly transduced subsets as well as nontransduced control cells for RNA-seq. We focused on genes differentially expressed between NLTB doubly (G+C+ and nontransduced (G−C−) cells from the earliest sets of cultures (days 14 and 24), and identified 468 differentially expressed genes (Fig. 5a, Supplementary Data 3). We performed Reactome pathway analysis[22] and found pathways relating to HOX genes in development/differentiation were second only to activated NOTCH signaling among top upregulated pathways (Supplementary Table 1), and immune/cytokine signaling multiply represented among the top downregulated pathways (Supplementary Table 2). The apparent coordinate regulation of multiple HOXB genes was notable and so we examined mRNA read counts across the entire HOXB locus and flanking regions. We found HOXB genes from B2 through B5 were significantly upregulated in NLTB G+C+ cells as compared to G−C− control cells (Fig. 5b). Of note, N-only (G+C−) cells showed variably significant, but overall intermediate levels of HOXB gene expression, suggesting that N alone can activate HOXB gene transcription, but that this effect is amplified in the context of LTB. Interestingly, we found that upregulation of multiple HOXB genes was unique to the combination of N + LTB, as it did not occur for N in combination with any of the other T-ALL oncogenes tested (LYL1, TLX1, TLX3, HOXA9, MEF2C, NKX2.1) (Supplementary Fig. 9). Thus, NLTB uniquely induces transcriptional activation of a contiguous cluster of anterior HOXB genes.

We also took the opportunity to compare RNA-seq data between NLTB-transduced CB cells in vitro and corresponding primary NLTB leukemias that had developed in vivo (Supplementary Data 4) to gain insight into transcriptional programs enacted during tumor evolution/progression. Reactome pathway analysis of 31 genes upregulated in vivo emphasized Notch and RAS/RAF/MAPK signaling (Supplementary Table 3), suggesting that growth in vivo selects for further enhancement of Notch signaling and corroborating our finding of acquired NRAS G12D

mutation from exome analysis (Fig. 4a). Pathways downregulated in vivo were dominated by interleukin/cytokine signaling (Supplementary Table 4), suggesting that cytokines/growth factors may be limiting in vivo, or perhaps that a substantial proportion of cells in vivo reside in less replete microenvironments. Alternatively, fully evolved leukemias may include subsets of cells that are less receptive to signaling agonists.

**NLTB induces altered epigenetic patterning over HOXB genes.** The contiguous distribution of upregulated genes within the anterior (3′) HOXB locus led us to wonder if altered epigenetic patterning might underlie the gene expression changes. Indeed, ChIP-seq analysis from the same set of samples showed significant H3K27me3 loss in G+C+ cells as compared to G−C− control cells at two regions, one near the 3′ end of HOXB2 and the other spanning from upstream of HOXB3/B4 to downstream of HOXB6 (regions 1 and 2, respectively, in Fig. 5c). Region 2 was particularly notable as it showed a corresponding enrichment of H3K27ac marks in G+C+ over G−C− cells (Fig. 5d). Taken together, these results reveal that NLTB-transduced cells exhibit an altered chromatin pattern consistent with gene activation over the HOXB locus (corresponding RNA-seq tracks in Supplementary Fig. 10), and suggest that NLTB may initiate the leukemogenic process by remodeling of chromatin to achieve coordinate regulation of multiple genes required for cellular transformation.

**High HOXB is associated with poor clinical outcome in T-ALL.** To begin to address what role HOXB genes may play in natural/spontaneous human T-ALL, we examined RNA-seq data from a large pediatric cohort comprising 264 diagnostic T-ALL samples (COG TARGET study)[8]. HOXB2, B3, and B4 were expressed in a subset of cases, and in a notably coordinated fashion (Fig. 6a), corroborating mRNA expression data from transduced CB cells (Fig. 5b). HOXB5 was expressed in only a minority of cases, and was expressed at lower levels as compared to HOXB2/B3/B4 in transduced CB cells. Strikingly, cases with higher levels of HOXB2-B5 mRNA expression exhibited significantly poorer event-free survival (Fig. 6b, Supplementary Table 5). HOXB4 alone was also significant (Fig. 6c), while HOXB2 and HOXB3 were not significant (Fig. 6d, e). These data support that HOXB genes are coordinately expressed in primary human T-ALL and higher levels of expression are associated with more aggressive clinical disease.

**High HOXB defines a distinct disease subgroup in T-ALL.** The poorer clinical outcomes of patients with high HOXB gene expression led us to wonder if they might overlap with cases of ETP-ALL[23]. We found only limited overlap between these two groups by PCA (Fig. 7a) and no positive statistical association, but instead a trend towards negative association (Fig. 7b). We also examined whether high HOXB cases were enriched/depleted within any particular transcription factor/translocation-defined subgroup[8]. Interestingly, high HOXB cases were significantly associated with TAL1, NKX2-1, and so-called "unknown" subgroups, while low HOXB cases were significantly associated with TLX1 and TLX3 subgroups (Fig. 7c). Accordingly, HOXB2/B3/B4 gene expression levels were consistently elevated in TAL1, NKX2-1, and "unknown" subgroups, and consistently decreased in TLX1 and TLX3 subgroups (Supplementary Fig. 11). Of note, cases in the top quintile for expression of TAL1 show higher expression of HOXB2/B3/B4, although this difference was statistically significant only for HOXB4. There was no evidence for increased HOXB gene expression in the top quintile of LMO2 expressers

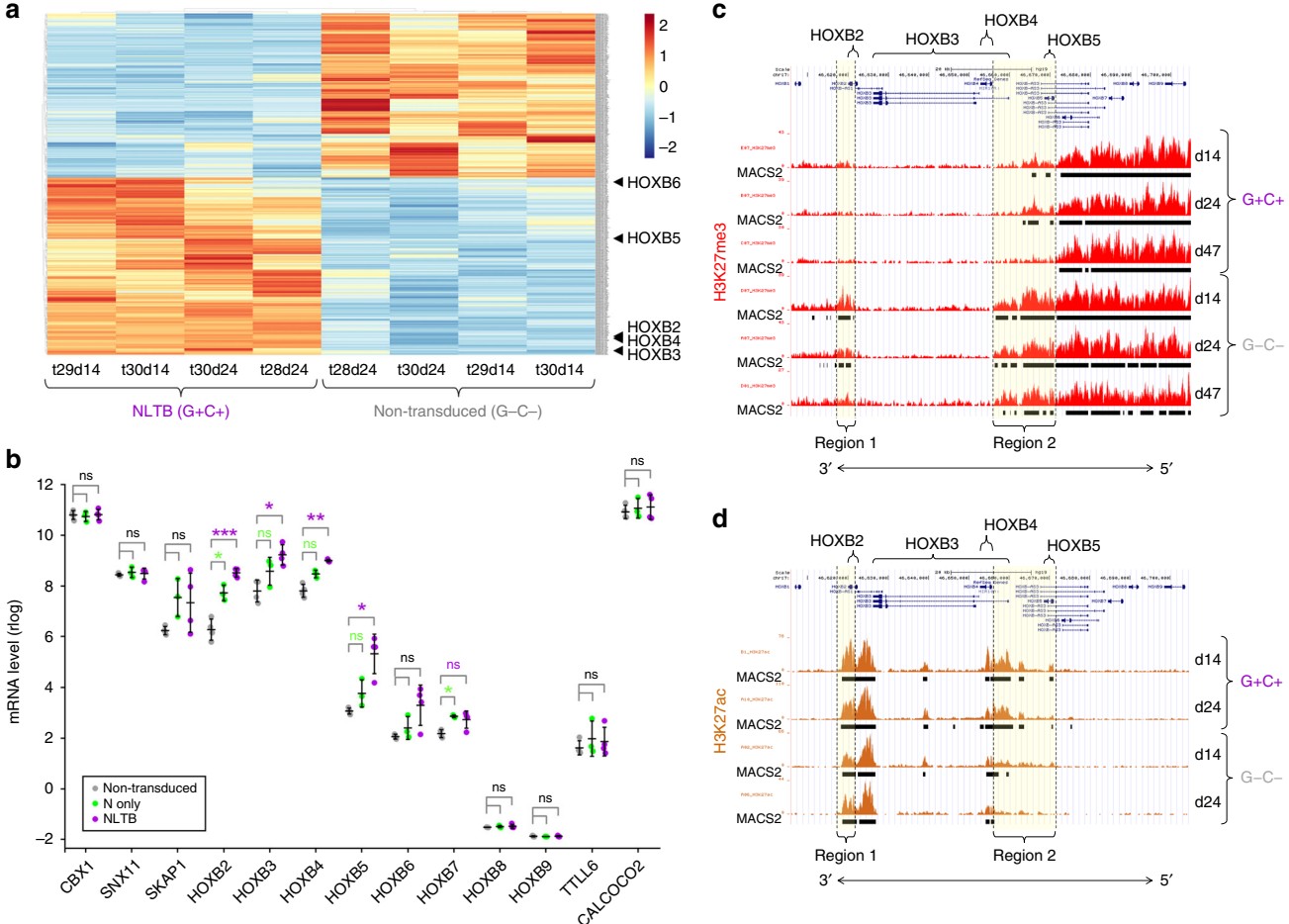

**Fig. 5** NLTB induces anterior HOXB gene expression with altered epigenetic patterning. **a**, **b** RNA-seq analysis of nascently transduced CB cells. CB cells were transduced with N-GFP and LTB-Cherry lentiviruses, then cultured on OP9-DL1 feeders for 14–24 days. RNA was prepared from FACS-sorted doubly (G+C+), singly (G+C−), and nontransduced (G−C−) subsets. Sorted G−C+ cells did not pass QC and are not shown. Samples were collected from three different experimental trials (t28–t30). **a** mRNA expression heatmap. Differentially expressed genes (G+C+ vs. G−C−; log2FC > 1, p-adj < 0.1) are depicted (468 total, 243 up/225 down). Heatmap is scaled by gene (=row) with mean = 0 and SD = 1. **b** mRNA expression level of HOXB and flanking genes. Rlog values from DESeq2 are plotted. Each datapoint represents an individual sample. Bars indicate mean ± SD for each set of colored dots. Statistical comparisons are for doubly vs. nontransduced cells. *p < 0.05; **p < 0.01; ***p < 0.001; ns not significant (two-tailed t test with Holm−Sidak correction for multiple comparisons). **c**, **d** Histone ChIP-seq tracks for **c** H3K27me3 and **d** H3K27ac over the HOXB locus in NLTB (G+C+) and nontransduced (G−C−) CB cells cultured for 14, 24, or 47 days on OP9-DL1 feeders. Subsets were FACS sorted prior to fixation for ChIP. Corresponding MACS2 peak calls are shown below each track

(Supplementary Fig. 12). Additional studies will be needed to assess what genetic contexts are most permissive to HOXB gene upregulation.

To explore further what features might characterize patient T-ALL cases with high vs. low HOXB gene expression, we performed gene expression enrichment analysis (GSEA) using gene signatures derived from an RNA-seq dataset that included normal hematopoietic progenitors from human bone marrow and thymus[24]. PCA suggested three groupings of normal cells: (1) HSC and LMPP subsets; (2) early thymic progenitors through DN stage; and (3) late thymic progenitors including DP, SP4, and SP8 subsets (Fig. 7d). Defining gene signatures by pairwise differentially expressed gene sets, we found high HOXB cases to show significant enrichment for the late thymic over early thymic, late thymic over HSC/LMPP, and early thymic over HSC/LMPP gene signatures (Fig. 7e). These findings describe high HOXB cases as bearing a gene signature of late thymic progenitors, and are associated with TAL1, NKX2-1, and "unknown" genetic subgroups. The association with multiple different genetic subgroups would suggest that commonality among high HOXB

cases is defined by a biologic process orthologous to conventional transcription factor groupings.

### HOXB3 promotes cell growth in pre- and established leukemia.
Following on evidence that HOXB gene upregulation occurred as an early event in NLTB-transduced CB cells, we examined RNA-seq data from established CB leukemias and a collection of T-ALL PDXs and found HOXB2/3/4 mRNA levels were coordinately elevated in both contexts (Fig. 8a). We also looked at HOXB gene expression in other acute leukemias and found that both T-ALL and B-ALL PDXs were characterized by specific elevation of anterior HOXB genes, while AML PDXs showed a pattern of HOXB gene expression that extended more broadly across the locus (Supplementary Fig. 13). Interestingly, levels of HOXB2/3/4 were significantly higher in T-ALL than in B-ALL, suggesting perhaps a particular relevance of anterior HOXB gene action in T-ALL.

To assess formally whether these HOXB genes play a functional role in established leukemia, we performed a limited scale, pooled shRNA dropout screen using primary CB leukemia cells cultured in vitro. The shRNA pool included a total of 56 shRNAs targeting

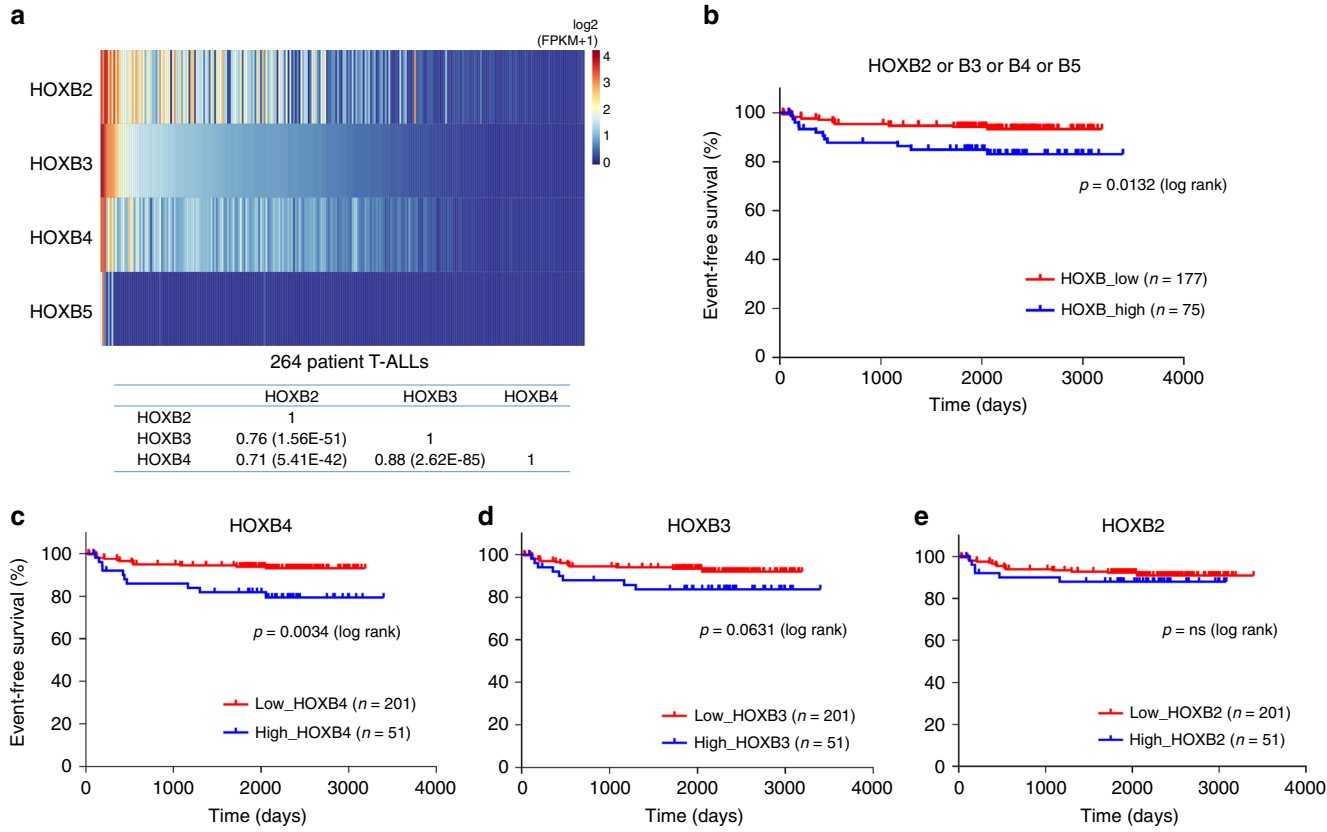

**Fig. 6** High HOXB gene expression is associated with poor clinical outcome in human T-ALL. Analysis of 264 patient T-ALLs from the COG TARGET study (dbGaP phs00018/000464). **a** mRNA expression heatmap and Pearson correlation matrix. HOXB mRNA levels from RNA-seq data. Color scale is based on log2(FPKM+1) values without further normalization. Samples are ordered left-to-right by decreasing HOXB3 level. Correlation $p$ values are indicated in parentheses. **b–e** Event-free survival among 252 T-ALL patients. Twelve patients were excluded for withdrawn consent, second malignancy, or were inevaluable. **b** HOXB_high group. Cases were defined as greater than 80th percentile mRNA level and FPKM > 1 for HOXB2, B3, B4, or B5. **c** HOXB4_high group. **d** HOXB3_high group. **e** HOXB2_high group. The HOXB5_high group included only three cases and thus was not plotted. ns not significant

multiple HOXB (2, 3, 4, 5) and HOXA (5, 7, 9, 10) genes (5–10 shRNAs per gene) plus three non-targeting controls (Supplementary Table 6 and Supplementary Fig. 14). We included four HOXA genes in the screen as they were significantly downregulated in G+C+ vs. G−C− cells (Supplementary Data 3), raising the possibility they may antagonize T-ALL growth. We found significant depletion of four different shRNAs against HOXB3 and two against HOXB5 (Fig. 8b; Supplementary Fig. 15), suggesting their knockdown had resulted in growth disadvantage. We also performed the same screen using established T-ALL cell lines HSB2 and PEER and found consistent depletion of shRNAs against HOXB3, but less consistent depletion of shRNAs against HOXB5 (Supplementary Fig. 16).

We focused further efforts on HOXB3 since HOXB5 expression was detectable in only a small number of primary T-ALLs (Fig. 6a). We confirmed accelerated depletion of three different HOXB3 shRNAs as compared to nonsilencing controls in singleton transduction experiments by flow cytometry using NGFR-tagged viral shRNA constructs in the same primary NLTB leukemia as used for the 59-plex shRNA screen, and also in a second, independent primary NLTB leukemia (Fig. 8c). Similar results were obtained using three different established human T-ALL cell lines (Fig. 8d) selected for study as they expressed relatively high levels of HOXB3 in combination with TAL1 and/or LMO2 (Supplementary Fig. 17). These findings are consistent with the notion that HOXB3 contributes to maintenance of established leukemias.

To address whether HOXB3 also played a role in the early stages of leukemia initiation, we went back to normal CB cells

nascently transduced with NLTB to test the effect of HOXB3 shRNAs on clonogenic cell growth. We employed a variation on conventional methylcellulose colony forming cell (CFC) assays, but compatible with OP9-DL1 co-cultures in which transduced CB cells were sorted at limiting dilution into individual wells of a 96-well plate containing OP9-DL1 feeders and then assayed ~3 weeks later by flow cytometry. Net yields per well of viable, triply transduced hCD45+ cells (N/GFP+, LTB/Cherry+, shRNA/NGFR+) were significantly decreased for shHOXB3_644 down to ten input cells per well and for shHOXB3_643 down to 50 input cells per well as compared to shScr control (Fig. 8e). If we analyze the data according to a single-hit model/Poisson distribution where a positive well is defined as containing at least 500 triply transduced hCD45+ cells, we are able to calculate the frequency of well-initiating cells, or WIC. By this approach, we find the WIC frequency for control shRNA-transduced cells to be ~1/41 cells (95% CI 1/30–56 cells), while shHOXB3-transduced cells showed statistically significant, 6- to 12-fold lower WIC frequencies of ~1/240 cells (95% CI 1/140–430) for shHOXB3-643 and ~1/510 cells (95% CI 1/230–1100 cells) for shHOXB3-644 ($p = 6.9e{-}10$ and $1.6e{-}13$, respectively; chi-square test) (Fig. 8f). Similar, statistically significant differences were also observed using alternate cell yield cutoff values as low as 200. Importantly, the goodness of fit test for shScr cells supports that the null hypothesis/single-hit model is not rejected ($p = 0.78$; chi-square test). Taken together, these results support the notion that NLTB-induced HOXB3 gene expression is required for clonogenic expansion of T-progenitor

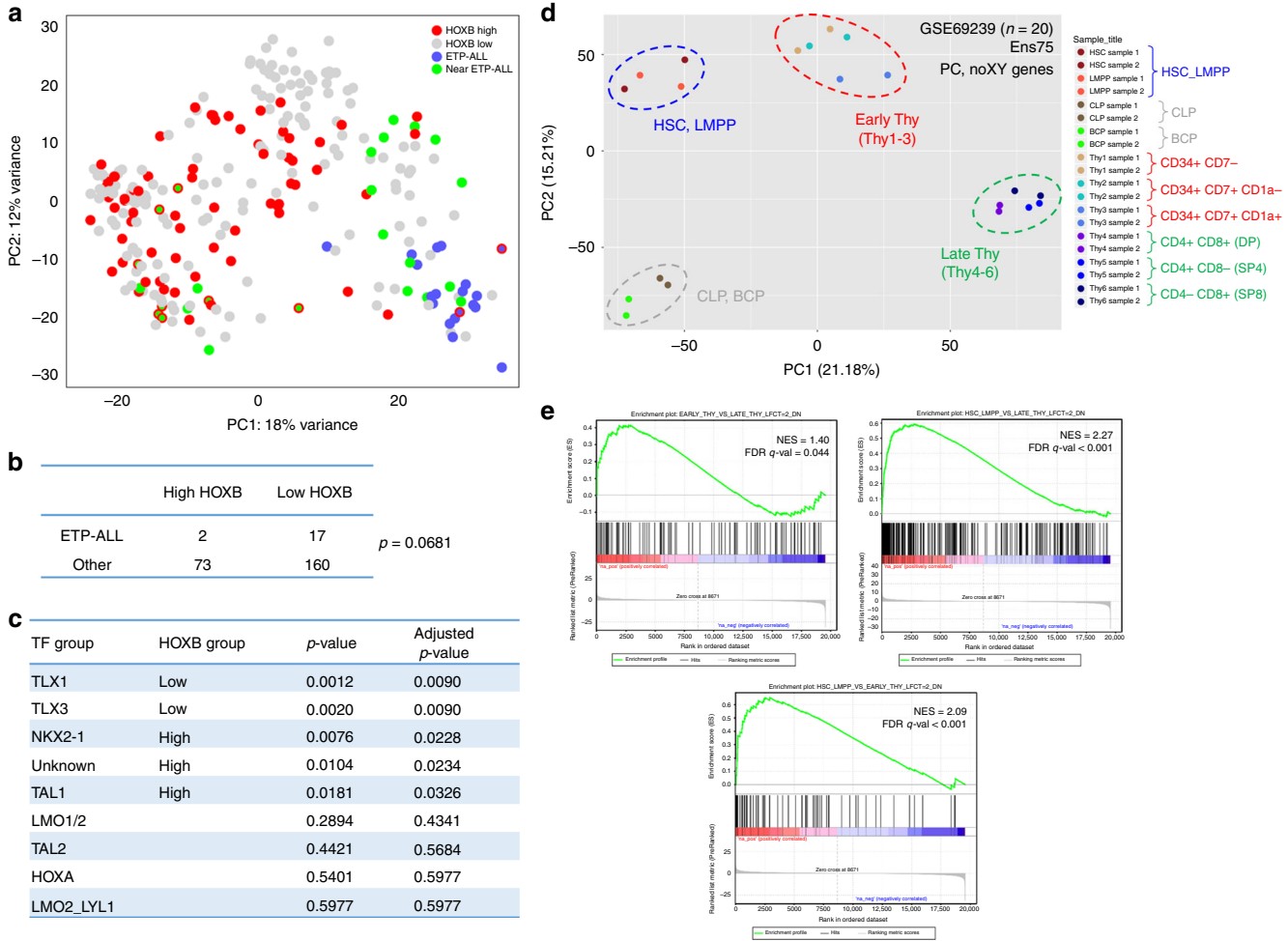

**Fig. 7** High HOXB gene expression defines a distinct disease subgroup in human T-ALL. **a** Principal component analysis (PCA) of RNA-seq data from 252 patient T-ALLs (COG TARGET study; dbGaP phs00018/000464). HOXB (high and low), ETP-ALL, and near ETP-ALL subgroups are highlighted. **b**, **c** Correlation between HOXB subgroups and (**b**) ETP-ALL or (**c**) transcription factor/genetic subgroups among 252 patient T-ALLs (COG TARGET study; dbGaP phs00018/000464). Statistical *p* values were calculated by Fisher's exact test and adjusted for multiple testing by the Benjamini−Hochberg procedure. **d** PCA of RNA-seq data from normal human bone marrow and thymic progenitor cells (GEO GSE69239). All protein coding genes (X/Y chromosomes excluded) were utilized for PCA. Samples were partitioned into four groups (HSC/LMPP, CLP/BCP, Early Thy = Thy1/2/3, Late Thy = Thy4/5/6) based on the PC1 and PC2 dimensions. **e** Gene set enrichment analysis (GSEA). Depicted gene signatures were derived by differential expression analysis of HSC/LMPP vs. early thymic vs. late thymic progenitors using a log2 fold-change (FC) threshold of 2. All protein coding genes (X/Y chromosomes excluded) were rank ordered according to the metric (sign(logFC)*(−log10(*p* value))) for differential expression in HOXB_high vs. HOXB_low patient T-ALLs from the COG TARGET study (*n* = 252; dbGaP phs00018/000464). Preranked GSEA was then run with 1000 permutations. NES normalized enrichment score, FDR false discovery rate

cells undergoing the earliest stages of malignant transformation, and that this dependency persists in established leukemia cells.

## Discussion

The synthetic leukemia model described here provides an efficient and reproducible means for generating human T-ALL that appears indistinguishable from spontaneously arising patient tumors. In addition to overcoming limitations associated with models involving transformation of mouse cells, it allows custom design/specification of the tumor's genetic composition, thus facilitating creation of near isogenic sets of tumor samples necessary to deconvolute the contributions of individual genetic elements to tumor phenotypes. This approach holds several advantages over deploying large collections of PDXs which contain a wide assortment of genetic variants that can obscure discovery/validation of bona fide genetic associations, and are logistically difficult to generate and share. As well, all tumors continue to evolve during serial propagation in vivo and

obtaining early passage vials from PDX lines can be difficult if not impossible. In contrast, synthetic leukemias can be generated again and again from normal cells, thereby resetting the cancer evolutionary clock.

The ability to achieve multilog expansion of transduced CB cells in vitro afforded us the unique opportunity to explore the earliest of biochemical events that occur as cells were redirected from normal to malignant developmental trajectories. We used this approach to uncover a pro-oncogenic role for anterior HOXB genes in preleukemia cells at a very early stage in their genesis. While there is substantial prior literature on the role of HOXA in human T-ALL[25–28] and AML[29], and even of HOXB in normal HSC expansion, this study emphasizes a functional role for anterior HOXB genes in human T-ALL not appreciated in several genome-wide efforts[8,30–32]. We might surmise that upregulation of HOXB genes may be omnipresent in human leukemia, and only revealed when silhouetted against a proximal normal counterpart without confounding effects of additional genetic hits.

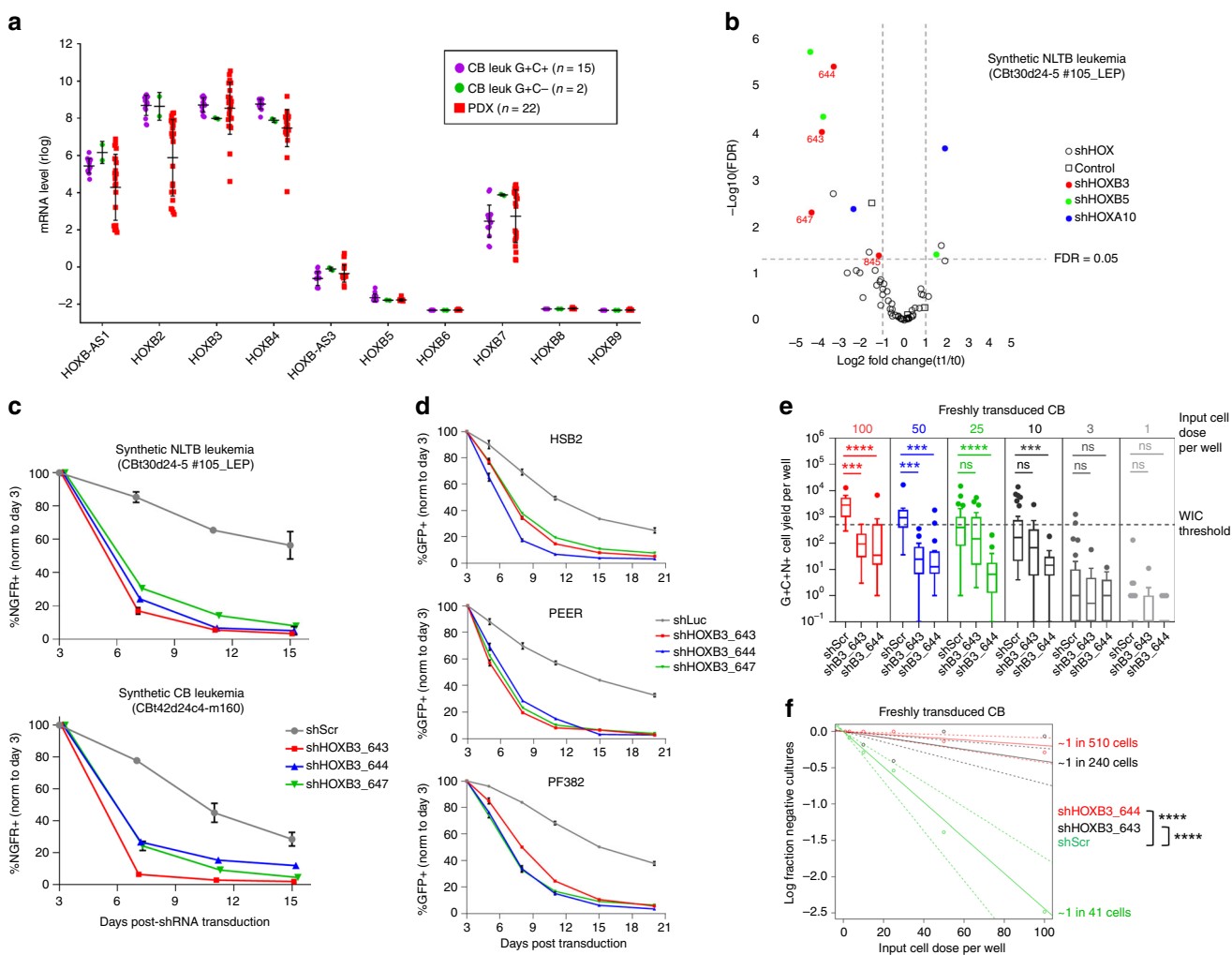

**Fig. 8** HOXB genes confer growth advantage in human T-ALL initiation and maintenance. **a** HOXB mRNA expression level in 17 CB leukemias and 22 PDX samples. Rlog values were calculated from RNA-seq data using DESeq2 and corrected for batch effects using ComBat. Each datapoint represents an individual sample with mean ± SD indicated. **b** Volcano plot of HOX shRNA depletion/enrichment. Primary CB leukemia cells were transduced with a 59-plex lentiviral shRNA pool targeting HOXA/B genes plus controls and cultured on OP9-DL1 feeders (two independent replicates; Spearman correlation 0.58). Genomic DNA was prepared from FACS-sorted cells on days 2 (t0) and 9/11 (t1) post transduction. Proviral shRNAs were PCR amplified and enumerated by NGS. Each datapoint represents a different shRNA species. Genes targeted by at least two different shRNA species with FDR < 0.05 are indicated in color. **c, d** Flow cytometric tracking of shRNA-transduced cell fraction over time in culture. **c** Primary NLTB CB leukemia cells and **d** human T-ALL cell lines. Mean ± SD fraction of the initial transduction value are plotted for experiments performed at least in triplicate. Decreasing NGFR+ or GFP+ fraction over time indicates shRNA-associated growth disadvantage. **e, f** Limiting dilution growth assays. CB cells were transduced with N + LTB viruses on days 0 and 4 and cultured until day 19, then transduced with shHOXB3 or control (shScr) lentiviruses and cultured until day 25. Triply transduced cells (N/GFP+, LTB/Cherry+, shRNA/NGFR+) were FACS sorted into individual wells of a 96-well plate containing OP9-DL1 feeders and cultured for ~3 weeks. The entire contents of each well were then harvested with trypsin and assayed by flow cytometry. **e** Total yields of viable, triply transduced cells per well ($n = 12–16$ for 50 and 100 cells per well; $n = 24–32$ for 25, 10, 3, and 1 cell(s) per well). Dotted line indicates threshold yield of 500 cells used for calculating well-initiating cell (WIC) frequencies. Box and whiskers are as defined by Tukey. ****$p < 0.0001$; ***$p < 0.001$; ns not significant (Kruskal–Wallis test with Dunn's correction for multiple comparisons). **f** WIC frequencies according to single-hit/Poisson model. Dotted lines indicate 95% CI. ****$p < 0.0001$ (chi-square test)

We had initially thought HOXB high cases might exhibit an HSC-like signature; however, our results suggest instead that these cases are more similar to late thymic progenitors. In combination with our functional data showing that HOXB3 and perhaps also HOXB5 support net cell growth in this context, we would posit that HOXB genes may be more accurately characterized as promoting expansion of lineage-restricted but immature progenitors, rather than the more prevalent view that they act to expand existing multipotent stem cells[33,34], an idea that has been proposed recently by others as well[35].

Exploration of the earliest events in cellular transformation may also favor identification of mechanisms operative in leukemia stem cells or that are shared by all cells in the tumor as opposed to later events that variably accumulate within individual subclones. Others have used variant allele frequencies to provide insight into the subclonal structure of tumors; however, this approach cannot inform on the cellular contexts in which variants occur or their associated phenotypes in isolation from subsequent mutational events. The synthetic approach affords an experimentally tractable way to model both the complement of genetic alterations and the cellular contexts in which they may occur.

Synthetic modeling offers a straightforward approach to deconvoluting the contributions of individual genetic

components in the leukemogenic process. For instance, we delivered NOTCH1ΔE and LTB on two separate viruses and thus could readily discern from in vitro cultures that NOTCH1ΔE provided an initial growth advantage, whereas LTB caused developmental arrest at an early stage with no apparent growth advantage. Further, while both NLTB and N-only produced aggressive leukemia in primary recipients, NLTB leukemias were consistently transplantable whereas most N-only leukemias were not transplantable. We interpret this to indicate that NOTCH1ΔE provides for prodigious expansion of immature T cells (enough even to overwhelm the primary recipient mouse), but in itself does not confer self-renewal properties required for serial trans-plantability. It is notable that in transplants using unsorted cells, G+C−/N-only cells always outnumbered G+C+/NLTB cells by ~6-fold on average, and yet G+C+/NLTB cells prevailed in over half of resulting leukemias. This supports the notion that LTB contributes positively to the leukemogenic process, perhaps even specifically to self-renewal. As further studies will be needed to identify what additional hits may be consistently present, we regard NLTB not as sufficient for leukemogenesis, but rather as sufficiently enabling to yield transplantable disease with high penetrance and reasonable latency. Of note, a similar CB T-ALL model was recently reported which utilized activated NOTCH1 only[36]; however, it was unclear how many independent trans-plantable clones were generated in that study, and according to our data, NOTCH1ΔE only does not reproducibly yield trans-plantable disease.

One limitation of our synthetic model is expression of onco-genes from viral elements; however, primary CB cells are amen-able to CRISPR/Cas9-mediated gene editing with targeting efficiencies ~80% reported in the literature[37], thus allowing more refined means of specifying the complement of genetic altera-tions. Another potentially limiting aspect is the in vitro culture phase prior to transplantation which may select for features dissimliar to propagation in vivo; however, the high level of similarity by RNA-seq between PDX samples and synthetic CB leukemias generated by transplantation after 10–24 days on OP9-DL1 feeders (Fig. 4b) suggests that untoward effects of culture in vitro may be limited. In counterbalance, the in vitro culture phase enables access to cells for biochemical and phenotypic assays that are not feasible otherwise, and thus represents a necessary evil if the earliest of events in cellular transformation are to be examined. Observations made in vitro will of course need to be verified using exclusively in vivo models and in pri-mary patient samples.

## Methods

**Isolation of human hematopoietic stem/progenitor cells.** Anonymized normal human cord blood (CB) samples were obtained with informed consent from women undergoing caesarian deliveries of full-term births according to protocols approved by the Research Ethics Board of the University of British Columbia and Children's & Women's Hospital of BC. CD34+ CB cells were obtained at >95% purity from pooled collections using a two-step Rosette-Sep/EasySep human CD34-positive selection kit (StemCell Technologies) according to the manufacturer's protocols and/or FACS sorting. The purity of FACS-sorted cells was >99% as determined by post-sort ana-lysis. CD34+ cells were seeded into 96-well round bottom plates and prestimulated in StemSpan SFEM II (StemCell Technologies) with 100 ng mL$^{-1}$ human SCF, 100 ng mL$^{-1}$ human FLT3L, 50 ng mL$^{-1}$ human TPO, and 20 µg mL$^{-1}$ human LDL, or 10 ng mL$^{-1}$ human SCF, 20 ng mL$^{-1}$ human TPO, 20 ng mL$^{-1}$ human IGF2, and 10 ng mL$^{-1}$ human FGFa (Peprotech) for 16 h.

**Lentiviral constructs and transduction.** Human NOTCH1 (ΔE allele), TAL1, BMI1, and mouse LMO2 cDNAs obtained from Dr. J Aster (Boston), Harvard PlasmID, and Dr. E. Lawlor (UCLA). NOTCH1ΔE and GFP cDNAs were con-nected with equine rhinitis A virus 2A (E2A) peptide[14]. LMO2, TAL1, BMI1 and Cherry cDNAs were similarly connected with thosea asigna virus 2A (T2A), foot-and-mouse disease virus 2A (F2A), and E2A peptides, respectively. These poly-cistronic cDNAs were cloned into pRRL-cPPT/CTS-MNDU3-PGK-GFP-WPRE[38]

immediately downstream of the MNDU3 promoter. All constructs were verified by sequencing. Additional vector construction details are available upon request.

High-titer lentiviral supernatants were produced by transient transfection of 293T cells using polyethyleneimine HCl MAX (Polysciences) with second-generation packaging/envelop vectors pCMV dR8.74 (Addgene #22036), pRSV-Rev (Addgene #12253), and pCMV VSV-G (Addgene #8454), followed by ultracentrifugal concentration (25,000 rpm for 90 min at 4 °C; Beckman SW32Ti rotor).

CB cells were transduced in 96-well plates coated with 5 µg/cm$^2$ fibronectin (StemCell Technologies) by direct addition of concentrated viral supernatants and transferred to OP9-DL1 co-cultures 6 h later. CB transduction efficiencies are shown in Fig. 1b, c, and Supplementary Data 1. CB leukemia and cell lines were transduced by spinoculation (1800 × g for 2 h at 33 °C) with viral supernatants in 4 µg mL$^{-1}$ polybrene. Primary CB leukemia transduction efficiencies were 15–30% for 105_LEP and 10–15% for m160. Cell line transduction efficiencies were 20–35% for HSB2, 20–30% for PEER, and 55–60% for PF382.

**Cell culture.** Transduced CB cells and explanted primary CB leukemia cells were cultured on top of confluent monolayers of OP9-DL1 cells in αMEM media sup-plemented with 20% FBS (Invitrogen) plus 10 ng mL$^{-1}$ SCF, 5 ng mL$^{-1}$ FLT3L and 3 ng mL$^{-1}$ IL-7 (Peprotech). Human T-ALL cell lines were cultured in RPMI 1640 media supplemented with 10% FCS, with 1 mM sodium pyruvate, 2 mM Gluta-MAX (Gibco), and antibiotics.

**Cell lines.** OP9-DL1 cells were obtained from J.C. Zuniga-Pflucker (University of Toronto). 293T, HSB2, PEER, and PF382 cells were obtained from J. Aster (Brigham & Women's Hospital, Boston). Cell line authentication by PowerPlex 16HS multiplex STR DNA profiling (Promega) was performed by Genetica DNA Laboratories (Burlington, NC). Cultures were confirmed mycoplasma-free and regular surveillance testing performed using the MycoAlert mycoplasma detection kit (Lonza).

**Patient-derived xenograft samples.** PDX samples from the PRoXe repository[6] are identified by "TA" prefix ($n = 17$). PDX samples established in the Weng lab include H3255-4, M22-7, M69-26, M71-1-2, and M30-2v2, and have been reported previously[39,40].

**Mice.** NSG (NOD.Cg-$Prkdc^{scid}$ $Il2rg^{tm1Wjl}$/SzJ) mice were bred and housed in a specific pathogen-free animal facility at the British Columbia Cancer Research Centre. All experimental procedures were approved by the University of British Columbia Animal Care Committee.

**Transplantation by intrahepatic/intravenous injection.** Neonatal NSG mice (4–10 days of age) were sublethally irradiated (100 cGy X-ray at 150 cGy min$^{-1}$), then injected intrahepatically with ~0.1 to 1.0 × 10$^6$ sorted CB cells (hCD45+ or hCD45+ GFP+ Cherry+) mixed with rhIL-7 (0.5 µg per mouse; Peprotech) and anti-IL-7 mAb (2.5 µg per mouse; clone M25; Bio X Cell, West Lebanon, NH) in PBS (total volume 30 µL). Mice were boosted with IL-7/IL-7 mAb cocktail by IP injection every 4–5 days for the first 28 days post transplantation. Adult NSG recipients were sublethally irradiated (200 cGy X-ray at 150 cGy min$^{-1}$) prior to intravenous injection of transduced CB cells (primary recipients) or CB leukemias (secondary recipients). Transplanted animals were monitored by monthly per-ipheral blood (PB) sampling and sacrificed at predefined, humane clinical mor-bidity endpoints.

**Histology.** Mouse tissues (spleen, thymus, lymph node, sternum) were fixed in 10% neutral-buffered formalin for 48 h, then stored in 70% ethanol before paraffin embedding. Hematoxylin and eosin staining was performed on 4 µm paraffin sections.

**Flow cytometry.** Absolute cell counts were obtained in flow data using AccuCheck counting beads (Invitrogen). Live/dead cell gating was performed by staining with propidium iodide or DAPI (Invitrogen). We used anti-hCD271 (Miltenyi Biotec, Biolegend) to detect the lentiviral NGFR marker. We performed flow cytometric analyses on FACSCalibur and LSRFortessa instruments and sorting on FACSAria3 and Fusion instruments (BD Biosciences). We analyzed flow cytometry data using FlowJo software (TreeStar). Example gating strategies are provided in Supple-mentary Figs. 18 and 19.

**Western blot.** Whole-cell lysates were separated by SDS-PAGE, transferred to Hybond-ECL membranes (Amersham) and blocked with 5% nonfat dry milk. Membranes were probed with primary antibodies the FLAG epitope (M2 clone; Sigma) or β-actin (AC-15; Sigma Aldrich), then with HRP-conjugated secondary antibodies (Jackson ImmunoLaboratories) and detected with enhanced chemilu-minescence (ECL; Pierce). Band intensities were quantified with Image Studio Lite (LI-COR) software.

**BIOMED-2 TCRG assay**. The BIOMED-2 TRG gene clonality assay for ABI fluorescence detection (InVivoScribe) was performed by the BC Cancer Agency Cancer Genetics Lab.

**ImmunoSEQ assay**. Genomic DNA was prepared from FACS-sorted cells using Qiagen AllPrep DNA/RNA micro kit according to the manufacturer's instructions. Samples were quantified using Nanodrop, diluted for library preparation in buffer EB, and submitted to Adaptive Biotechnologies (Seattle, WA) for immunoSEQ TCRG assay (survey resolution). Briefly, somatically rearranged human TCRG CDR3 was amplified from genomic DNA using a two-step, amplification bias-controlled multiplex PCR approach[41,42]. The first PCR consists of forward and reverse amplification primers specific for every V and J gene segment, and amplifies the hypervariable complementarity-determining region 3 (CDR3) of the immune receptor locus. The second PCR adds a proprietary barcode sequence and Illumina® adapter sequences[43]. CDR3 libraries were sequenced on an Illumina instrument according to the manufacturer's instructions. Raw sequence reads were demultiplexed according to Adaptive's proprietary barcode sequences. Demultiplexed reads were then further processed to: remove adapter and primer sequences; identify and correct for technical errors introduced through PCR and sequencing; and remove primer dimer, germline and other contaminant sequences. The data are filtered and clustered using both the relative frequency ratio between similar clones and a modified nearest-neighbor algorithm, to merge closely related sequences. The resulting sequences were sufficient to allow annotation of the V(N)D(N)J genes constituting each unique CDR3 and the translation of the encoded CDR3 amino acid sequence. V, D and J gene definitions were based on annotation in accordance with the IMGT database (www.imgt.org). The set of observed biological TCRG CDR3 sequences were normalized to correct for residual multiplex PCR amplification bias and quantified against a set of synthetic TCRG CDR3 sequence analogs[41]. Data were analyzed using the immunoSEQ Analyzer toolset.

**Whole exome sequencing**. Genomic DNA was prepared from FACS-sorted cells using Qiagen AllPrep DNA/RNA mini kit according to the manufacturer's instructions. The SureSelectXT Human All Exon V5 predesigned capture library (Agilent #5190-6208) was used along with the SureSelect XT Library Prep Kit ILM (Agilent #5500-0132) in order to generate human exome libraries from gDNA as per the manufacturer's instructions. Libraries were paired-end 125 bp sequenced on an Illumina HiSeq 2500 at seven samples per lane. Reads were aligned to the human reference genome (hg19) using bwa-mem version 0.7.5a[44] with optical and PCR duplicates removed using the Picard tool (http://broadinstitute.github.io/picard/). Somatic SNV/indel variants were identified by VarScan[45] and were filtered for a minimum allele frequency of 1% and ten variant reads. Putative germline variants were removed based on a GMAF > 1%. All variants were annotated using SnpEff (version 4.2)[46] and filtered for effects predicted to have a moderate or high impact at the protein level.

**RNA-seq**. Total RNA was isolated from live cells with TRIzol reagent followed by purification over PureLink RNA Mini Kit columns (Invitrogen). RNA-seq was performed using a polyA-enriched or ribosomal RNA-depleted (NEBNext rRNA Depletion Kit; New England Biolabs, cat# E6310) strand-specific library construction protocol and paired-end 125 bp or 75 bp sequencing on an Illumina HiSeq 2500 instrument at eight samples per lane. Paired-end reads were trimmed from the 3′ end based on quality score (end min quality level (Phred) = 20) using Partek Flow software (version 6.0.17.0503; Partek Inc, St Louis, MO). Samples were then aligned to human genome reference assembly GRCh37/hg19 using STAR 2.5.2b[47] in Partek Flow software. All further analysis steps were done in RStudio 0.99.903 (R version 3.3.1). Raw gene expression counts were calculated using featureCounts in Rsubread v1.24.2 with GRCh37 Ensemble release 75 as annotation[48]. Genes with summated counts of 1 or less across all included samples were filtered out. mRNA expression values were then derived after normalization with rlog function in DESeq2 v1.18.1, which log2 transforms and normalizes the data for library size, as well as minimizes the effect of low-expression genes[49]. Differential gene expression analyses were performed using filtered raw counts with DESeq2 v1.18.1.

To remove batch effects between PDX and synthetic CB leukemia samples, we used ComBat function in sea 3.20.0 with nonparametric adjustments and specified biological covariates[50,51]. To determine relationships between PDX and synthetic CB leukemia samples, the top 1000 variable genes across PDX samples only were selected using genefilter v1.56.0 rowVars function[52], and Spearman correlation for all samples was calculated based on this gene list. Hierarchical clustering with complete linkage and distance measured as (1 − Spearman correlation) was performed and plotted with pheatmap 1.0.8 (https://CRAN.R-project.org/package=pheatmap). Multiscale bootstrap resampling was performed using pvclust in R (http://stat.sys.i.kyoto-u.ac.jp/prog/pvclust/).

**Native histone ChIP-seq**. We performed native chromatin immunoprecipitation (ChIP) using validated antibodies against H3K27me3 and H3K27ac and constructed ChIP-seq libraries according to established protocols[53,54]. Libraries were sequenced on an Illumina HiSeq 2500. Raw sequences were inspected for quality, sample swap, and reagent contamination using custom in-house scripts. Paired-end 125 bp reads were aligned to human genome build GRCh37-lite using BWA v0.5.7[55] and the alignment files were converted to bam format through SAM-tools[56]. Wig files were converted from the bam files via custom program BAM2-WIG (http://www.epigenomes.ca/tools.htm), excluding reads that were unmapped, duplicate, or having mapping quality score <5. Wig files were subsequently converted into big wig format for visualization in UCSC genome browser. Peak calling was performed using MACS2 (Model-based Analysis for ChIP-Seq, v2.1.1)[57] by pairing the IP bam and the corresponding input bam, with $q$ value cutoff = 0.01 for narrow histone mark (H3K27ac), and 0.1 (in broad mode) for broad histone mark (H3K27me3).

**shRNA library construction and transduction**. Separate 5 mL bacterial overnight cultures were grown for each of 56 different lentiviral shRNA constructs targeting a total of eight genes (average seven shRNAs per gene) (Supplementary Table 6), which were then pooled for a single plasmid DNA prep (Qiagen). The final plasmid prep was sequenced on an Illumina MiSeq to assess representation of each shRNA species. All input shRNA clones were detected in the final plasmid prep, with ~55% within 4-fold and ~75% within 10-fold of the mean read counts (Supplementary Fig. 14).

Lentiviral preps for the 56-plex pooled shRNA library and three negative control shRNAs (shScramble, shLuc, empty vector) were done separately and then mixed prior to transducing cells. Cells were transduced at a target multiplicity of infection of 0.3 to favor single lentiviral integration in the majority of transduced cells.

**shRNA knockdown growth screen**. The lentiviral shRNA vector, pLKO contains a puromycin selection cassette; however, we did not apply puromycin selection after transduction in order to minimize the time duration between initial shRNA transduction and collection of the t0 time point sample. Since cultures thus contained both transduced and nontransduced cells, an excess of cells was carried in culture for the duration of the screen in order to maintain at least ~60,000 transduced cells, corresponding to ~1000-fold representation of each shRNA clone. Cultures were passaged at regular intervals as needed to maintain logarithmic growth phase for a total of ~5 population doublings.

**shRNA enumeration by NGS**. shRNA hairpins were PCR amplified from DNAzol extracted genomic DNA (Thermo Fisher) according to protocols developed by the RNAi Consortium (GPP Web Portal). Briefly, we used PCR primers mapping to a region of the U6 promoter directly upstream of the shRNA sequence and that included P5 and P7 attachment sequences for binding to the Illumina flow cell, Illumina primer binding sites, a 6-nucleotide barcode sequence for library multiplexing, and a staggered region to create sequence diversity during cluster identification (Supplementary Table 7). In order to maintain library representation throughout the PCR amplification process, we processed all genomic DNA from each sample. No more than 825 ng of total DNA was included per reaction[58], corresponding to at least 500 templates per shRNA species (59-plex library, assuming 25% transduction efficiency). We performed a total of 30 cycles of amplification with Q5 High-Fidelity 2X Master Mix (New England Biolabs). Amplicons of the predicted size (300 bp) were gel purified from precast E-Gel EX 2% agarose gels (Invitrogen) using QIAquick gel extraction kit (Qiagen). The size and concentration of the each of the final libraries was verified by Agilent Bioanalyzer prior to pooling for sequencing on an Illumina MiSeq instrument. Paired-end FASTQ files from t0 and t1 were aligned to a reference list of shRNA hairpin sequences with 0 mismatches allowed. Differential shRNA representation was analyzed using the edgeR package in Bioconductor[59]. Results were plotted using the ggplot2 package in R. PCR primers and amplification conditions are available upon request.

**Statistics**. Quantitative data were analyzed using GraphPad Prism 8.0.1 software and various R packages. Well-initiating cell (WIC) frequencies were calculated from limiting dilution culture results using the online ELDA tool available at http://bioinf.wehi.edu.au/software/elda/ [60].

**Reporting summary**. Further information on research design is available in the Nature Research Reporting Summary linked to this article.

## Data availability

RNA-seq data from 17 of 22 T-ALL, 24 B-ALL, and 24 AML PDX samples referenced during the study are available in the NCBI SRA database under the accession code SRP103099. RNA-seq data and associated clinical annotations for samples from the COG TARGET study referenced during the study are available in the database of Genotypes and Phenotypes (dbGaP) under the accession code phs000218/000464. RNA-seq data for normal hematopoietic progenitors referenced during the study are available in the National Center for Biotechnology Information Gene Expression Omnibus (GEO) database under the accession code GSE69239. RNA-seq data for T-ALL cell lines referenced during the study are available in the European Genome-phenome Archive (EGA) database under the accession code EGAS00001000536. Whole exome sequencing (WES), RNA-seq, and ChIP-seqdata generated during the current study excluding that in Supplementary Fig. 9 have been deposited in the EGA database under accession code EGAS00001003627. ChIP-seq peak call (BED) files have been deposited in the GEO

database under accession code GSE130743. SNV calls from WES data underlying Fig. 4a are provided as Supplementary Data 5. Gene expression values from RNA-seq data underlying Figs. 4b/8a, 5a/b, and Supplementary Fig. 8 are included as Supplementary Data 6–8, respectively. Data from Supplementary Fig. 9 are available from the corresponding author upon reasonable request. All other data supporting the findings of this study are available within the article and its Supplementary Information files, or from the corresponding author upon reasonable request. A reporting summary for this article is available as a Supplementary Information file.

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

## Acknowledgements

This work was supported by Program Project Grant funding from Terry Fox Research Institute. M.K. received postdoctoral support from Japan Society for the Promotion of Science. A.C.S. received a Canada Graduate Scholarship-Master's studentship from Canadian Institutes of Health Research.

## Author contributions

M.K. and A.P.W. designed experiments, interpreted results, and wrote the manuscript; M.K., A.C.S., K.T., R.W., A.N., C. Shanna, S.G., E.A.C. and A.L. generated data; K.T., A.Z., A.H., S.B., R.A.H., S.H. and A.B. performed informatics analyses; X.W., C. Steidl, A.K., R.K.H., C.J.E. and M.H. provided advice and discussion.

## Additional information

**Competing interests:** The authors declare no competing interests.

