## [Peer Review File · Nature Communications]

Reviewers' comments:

Reviewer #1 (Remarks to the Author):

The manuscript by Kusakabe and coworkers describes the generation de novo of human T-cell leukemia from normal CD34+ cord blood (CB) progenitors transduced with a combination of known T-ALL oncogenes. Using this model, the authors provide evidence for oncogene-induced HOXB activation in established CB leukemia and show that elevated HOXB gene expression defines a novel subset of patients with poor clinical outcome. Finally, they demonstrate that anterior HOXB genes are specifically activated in human T-ALLs by an epigenetic mechanism and conclude that this mechanism may confer growth advantage in both pre-leukemia cells and established clones.

The work is well performed and provides interesting results on the activation of anterior HOXB genes in de novo established CB leukemia whose clinical relevance is supported by the finding of a similar gene activation pattern in T-ALL patients. However, no functional data are provided to formally demonstrate that HOXB genes play a key pro-oncogenic role at the very early stages of leukemia generation, as the authors claim. In addition, the complexity of the leukemia model, involving the overexpression of several T-ALL oncogenes together with an in vitro-expansion step prior to transplantation, may represent important limitations to establish the contribution of individual genetic components to the leukemic process in patients.

Major concerns:

The generation of leukemia from in vitro-expanded CB cells raises the question of whether leukemia is generated in vivo from pre-leukemic cells expanded in vitro, or leukemic clones are generated and selected in vitro prior to transplantation. This is an important issue that needs further clarification. The authors should at least analyze whether in vitro cells that generate leukemia in vivo display clonal TCR gene rearrangements.

It is shown that there is in vitro and in vivo selection for G+C+ over G+C- cells. However, G+C- N-only leukemias can be generated in vivo and are transplantable in about 50% of secondary recipients. Based on these results the authors conclude that N alone yielded lethal, yet non-self renewing lymphoid expansions in about half of instances. An alternative possibility is that both N-only and N+LTB combinations induce self-renewing lymphoid expansions in vivo, but with dissimilar

efficiencies. Therefore, quantitative data on the frequencies of leukemia-initiating cells (LICs) in sorted G+C⁻ and G+C⁺ transplanted leukemias should be provided.

In order to assess the nature of the very first molecular changes underlying malignant transformation of normal CB cells, the authors isolate transduced and non-transduced cell subsets from cultures and perform RNAseq and ChIPseq studies. Although very informative, these results do not define the timing of genetic alterations. A comparison of the in vitro and in vivo leukemic counterparts should be provided.

Based on results in Fig. 4, the authors conclude that NLTB may initiate the leukemogenic process by remodeling chromatin to achieve coordinate regulation of multiple genes required for cellular transformation, particularly of HOXB genes. The possibility that activated NOTCH signaling is sufficient to induce the process is not assessed and cannot be formally excluded. At least, statistical comparisons of N-only and non-transduced cell samples shown in Fig. 4b should be provided. In this regard, how do the authors explain that high HOXB T-ALL cases, like N-only leukemias, display a late thymic progenitor phenotype, while NLTB leukemias with elevated HOXB activation are similar to early progenitors?

In Figure 7c,d,e, proportions of shScr-transduced control cells decrease along culture, suggesting a non-specific impact of the shRNA transduction procedure on in vitro cell proliferation. This is particularly important regarding Fig. 7e data, which may support the claim that HOXB genes are crucial for the initiation of leukemia. Thus, formal proof should be provided that freshly transduced CB control cells in Fig. 7e do actually generate leukemia in contrast to shHOXB3-transduced cells.

Minor points:

In Figure S3, both G+C⁺ and G+C⁻ leukemias with a DP TCR α phenotype are shown. How do the authors explain that most of these cells are negative for CD3?

Why were transplanted mice boosted with IL-7/IL-7 mAb? What is the impact of IL-7 on leukemia generation/progression?

Reviewer #2 (Remarks to the Author):

Kusakabe, et al describe the generation of synthetic T-cell acute lymphoblastic leukemia by lentiviral transduction of human hematopoietic cells. The authors show that the leukemias that develop phenotypically recapitulate naturally occurring T cell ALL. They go on to use their synthetic model to identify aberrant anterior HOXB gene expression in their model and correlate with primary T ALL sample data. The model system described here is of interest, as creating genetic models of leukemia in human cells has obvious advantages over use of other model systems such as murine models, PDX models and human cell lines. In general the paper is well written and the figures are of high quality. However, a few issues should be addressed before the paper is suitable for publication.

1. The authors only present data from N+LTB transduction, but other combinations of were also performed. Do these other synthetic leukemias similarly relatively overexpress anterior HOXB cluster genes? If so rather than be specific to LTB transduction, perhaps this pattern of expression is just inherent to synthetic T cell leukemia derived from cord blood stem/progenitor cells.

The authors report that HOXB overexpressing T ALL patient samples were associated with TAL1, NKX2-1 and "unknown". But if the authors look instead at TAL1/LMO2 cases, is there consistent overexpression of anterior HOXB genes?

2. The exact methodologies employed by the authors in various experiments is confusing to follow. They state that for early protocols cells were harvested from OP9 culture at 10 days and transplanted into neonatal mice, but later experiments used sorted G+C+ cells harvested at 24-25 days. In figure 2, presumably mice from strategy 1 are depicted, but this is not made clear. Why were the methods changed? Presumably most of the data presented is from strategy 1 given leukemias of G+C+, G+C- and mixed populations were documented from the same sample as shown in figure 2a, and gene expression from these differing types of leukemia were used to generate gene expression analyses comparing G+C+ to G+C- leukemias.

3. In figure 1e, the legend states that the plotted data are from either 19 or 33 days in culture post transduction. Please indicate which points are from 19 and which from 33. Are these from the same samples, analyzed at two time points? If so, is there any change in immunophenotype over time?

4. In figure 3a, the legends states that there were STR patterns from 4 different individual donors, whereas in the text it states in line 158 that 'at least 5 different STR profiles' were present. Please clarify.
5. In figure 4, it would be helpful to show the corresponding RNAseq tracks.
6. In figure 5C and 5D, it appears there were 51 patients with high HOXB3 expression and 51 with high HOXB4 expression. Is this just coincidentally the same number of patients (some presumably the same, but some must differ because the curves are different)? Maybe this is the correct number for both, but just want to verify.
7. In supplemental figure S3B, please use an arrow to indicate the bottom row is gated G+C-.
8. In line 99, self-cleaving should technically be in quotations, since the process is ribosome skipping, not actual self-cleavage.
9. Suggest avoiding the use of the phrase "just missed the $p < 0.05$ cutoff" in line 246. Rather just state it was not statistically significant.
10. Rather than "packed" bone marrow in line 128, I suggest the authors use a more objective description of the bone marrow morphology such as hypercellular or infiltrated with leukemic blasts, for example.

Reviewer #3 (Remarks to the Author):

In this manuscript, Wang et al report a synthetic model of T-cell leukemia and discover the role of HOXB genes in its initiation and maintenance. More specifically, to show the functional importance of these factors, the authors perform a focused shRNA drop-out pooled screen in NLTB leukemia and T-ALL cell lines. The reviewer has several questions and/or suggestions regarding this part of the manuscript:

1. The methods section describing the screen is written with insufficient detail. Has the custom-made pooled library been sequenced to check for initial representation of the shRNA? If not, how did the T0 representation look like? How were the transduced cells selected for? Please state how many biological replicates were performed. What was the correlation between independent replicates?
2. The authors state in the main text that they conducted the same screen in T-ALL cell lines with HOXB3 showing consistent drop out. Please show volcano plots in a supplementary file.
3. In the primary screen, why was T1 collection point chosen to be at Days 9-11? Typically longer time intervals are needed for negative selection screens to achieve a better dynamic range. Are these cells proliferating much faster compared to T-ALL cell lines where it took longer for shHOXB3-transduced cells to drop out (e.g. Figure 7D)?
4. It appears that validation experiments in figure 7C are performed in another NLTB cell line compared to the one used in the primary screen. If this is correct, it would be important to show validation using the cell line from the screen. These screens are inherently noisy and validation is critical. Are the 3 shRNAs against HOXB3 scored as hits in the screen the same as the ones outlined in figure 7C, D and E?
5. As part of the validation protocol, it is also important to show that shRNA act on-target. A western blot and/or qPCR showing an extent of knock down using the constructs against HOXB3 should be shown in a supplementary file. It would be also interesting to check the performance of HOXB5 shRNA to eliminate potential lack of efficacy of these shRNAs.
6. Graphs in figures 7C, D, and E do not show any error bars. Please display error bars at least for technical replicates or if they were not performed state why not.

Reviewer #1 (Remarks to the Author):

The manuscript by Kusakabe and coworkers describes the generation de novo of human T-cell leukemia from normal CD34+ cord blood (CB) progenitors transduced with a combination of known T-ALL oncogenes. Using this model, the authors provide evidence for oncogene-induced HOXB activation in established CB leukemia and show that elevated HOXB gene expression defines a novel subset of patients with poor clinical outcome. Finally, they demonstrate that anterior HOXB genes are specifically activated in human T-ALLs by an epigenetic mechanism and conclude that this mechanism may confer growth advantage in both pre-leukemia cells and established clones.

The work is well performed and provides interesting results on the activation of anterior HOXB genes in de novo established CB leukemia whose clinical relevance is supported by the finding of a similar gene activation pattern in T-ALL patients. However, no functional data are provided to formally demonstrate that HOXB genes play a key pro-oncogenic role at the very early stages of leukemia generation, as the authors claim.

>> We have now performed additional functional assays to help substantiate the contribution of HOXB3 to the early stages of leukemia generation. Specifically, we have introduced a new “well initiating cell”, or WIC assay which scores the capacity of individual transduced CB cells to expand *in vitro*. We present this as a feeder-compatible variant of the more conventional methylcellulose colony forming cell (CFC) assay. We use this assay to show that knockdown of HOXB3 in individual NLTB cells significantly reduces their clonogenic potential (new Fig 8e/f). We feel this new data provides compelling evidence in support of the idea that HOXB3 contributes to clonogenic expansion at a very early stage in leukemia generation in our model.

In addition, the complexity of the leukemia model, involving the overexpression of several T-ALL oncogenes together with an *in vitro*-expansion step prior to transplantation, may represent important limitations to establish the contribution of individual genetic components to the leukemic process in patients.

>> We appreciate the limitations of our model with respect to the natural leukemic process that occurs in patients and have added text at the end of the discussion section acknowledging this important point.

Major concerns:

1) The generation of leukemia from *in vitro*-expanded CB cells raises the question of whether leukemia is generated *in vivo* from pre-leukemic cells expanded *in vitro*, or leukemic clones are generated and selected *in vitro* prior to transplantation. This is an important issue that needs

further clarification. The authors should at least analyze whether *in vitro* cells that generate leukemia *in vivo* display clonal TCR gene rearrangements.

>> This is an excellent point that we feel we can address in multiple ways. First, we would point out that leukemias arising in animals transplanted with the same pool of transduced CB cells exhibit distinct clonal TCRG rearrangements as assessed by a clinical-grade BIOMED-2 assay (**Fig 2c, 3b, S5**) and also distinct donor STR patterns (**Fig 3c, S6**), suggesting that a dominant, clonally rearranged leukemia had indeed not already arisen *in vitro* prior to transplantation. Second, we used the BIOMED-2 assay to assess if clonal TCRG gene rearrangements could be detected from *in vitro*-expanded CB cells. As shown in the new **Fig 3a**, there is no convincing evidence of dominant clonal TCRG rearrangements within *in vitro*-expanded CB cells from any of the 4 different CB trials tested. Further, comparison of TCRG profiles from *in vitro*-expanded CB cells and the leukemias they produced after transplantation reveals no clear evidence for perdurance of a dominant clone (**Fig 3b, S5**). Third, we submitted *in vitro*-expanded CB cells for commercial ImmunoSEQ TCRG assay (Adaptive Biotechnologies). We included NLTB-transduced CB cells (sorted G+C+) from days 14, 28, and 38 in culture, along with corresponding non-transduced control cells from the same cultures (sorted G-C-). Immunoseq TCRG analysis revealed normal Gaussian distributions of CDR3 fragment lengths for both transduced G+C+ cells and G-C- controls (**Fig S7**). Moreover, tracking of individual rearrangements showed no evidence of dominant clones emerging over time in culture among G+C+ cells, with the clone distribution pattern appearing highly similar to that of control G-C- cells (**Fig 3d**). Taken together, these results argue against the possibility that leukemic clones are generated and selected *in vitro* prior to transplantation.

2) It is shown that there is *in vitro* and *in vivo* selection for G+C+ over G+C- cells. However, G+C- N-only leukemias can be generated *in vivo* and are transplantable in about 50% of secondary recipients. Based on these results the authors conclude that N alone yielded lethal, yet non-self renewing lymphoid expansions in about half of instances. An alternative possibility is that both N-only and N+LTB combinations induce self-renewing lymphoid expansions *in vivo*, but with dissimilar efficiencies. Therefore, quantitative data on the frequencies of leukemia-initiating cells (LICs) in sorted G+C- and G+C+ transplanted leukemias should be provided.

>> We feel our wording on this point was unclear; in fact, we agree that both N only and NLTB can produce serially transplantable (ie self renewing) leukemias, but that the lower efficiency of this process in N only cells is likely due to the need for stochastic acquisition of additional oncogenic hits. Once these additional hits are acquired, we have no reason to suspect that the resulting nominally "N only" leukemias are any less transplantable than NLTB leukemias. The distinction we were attempting to emphasize was that whereas all assayed NTLB leukemias

were serially transplantable into secondary recipients (**Fig 2d**; 6/6 clones), less than half of assayed “N only” leukemias performed similarly (**Fig 2e**; 2/5 clones). We have amended the results text to clarify this point.

Nonetheless, we did still perform a limiting dilution transplant analysis experiment as requested from which we were able to calculate the LIC frequency of sorted G+C+ cells from an NLTB leukemia to be greater than 1 in 4,100 cells, as compared to the LIC frequency of sorted G+C- cells from a “non-transplantable” leukemia to be less than 1 in 1,200,000 cells (at least ~300-fold difference). This new LDA transplant data is presented in **Fig S4**. To reiterate, we felt it would not have been informative to perform an LDA experiment using sorted G+C- cells from a transplantable leukemia (**Fig 2e**; 2/5 clones) since these are presumably no longer “N only” leukemias due to spontaneous acquisition of additional, but as yet uncharacterized hits.

3) In order to assess the nature of the very first molecular changes underlying malignant transformation of normal CB cells, the authors isolate transduced and non-transduced cell subsets from cultures and perform RNAseq and ChIPseq studies. Although very informative, these results do not define the timing of genetic alterations. A comparison of the *in vitro* and *in vivo* leukemic counterparts should be provided.

>> We have now performed RNA-seq comparisons as requested between *in vitro*-expanded CB cells and resulting *in vivo* leukemias. Using $\log_2FC > 2.5$ and $FDR < 0.1$ cutoffs, we identified 96 genes expressed more highly *in vitro* and 31 genes expressed more highly *in vivo*. Reactome pathway analysis of genes upregulated *in vivo* emphasized Notch and RAS/RAF/MAPK signaling, suggesting that growth *in vivo* selects for further enhancement of Notch signaling and corroborating our finding of acquired NRAS G12D mutation from exome analysis (**orig Fig 3b, new Fig 4a**). Pathways downregulated *in vivo* were dominated by interleukin/cytokine signaling, suggesting that cytokines/growth factors may be limiting *in vivo*, or perhaps that a substantial proportion of cells *in vivo* reside in less replete microenvironments. Alternatively, fully evolved leukemia cells may subsets of cells that are less receptive to signaling agonists. We have included these data in the new **Tables S6-S8**.

4) Based on results in Fig. 4, the authors conclude that NLTB may initiate the leukemogenic process by remodeling chromatin to achieve coordinate regulation of multiple genes required for cellular transformation, particularly of HOXB genes. The possibility that activated NOTCH signaling is sufficient to induce the process is not assessed and cannot be formally excluded. At least, statistical comparisons of N-only and non-transduced cell samples shown in Fig. 4b should be provided.

>> In the revised **Fig 5b**, we now provide statistical comparisons of HOXB gene expression in N-only vs. non-transduced CB cells and found that while HOXB2 and HOXB7 were significantly increased, no other genes within the HOXB cluster showed significant change. Nonetheless, the pattern of data would suggest that N does indeed contribute to supporting HOXB gene expression, albeit to a lesser extent as compared to the combination of N+LTB. We have amended the text to reflect more clearly the notion that some component of the increase in HOXB gene expression is likely provided by N alone.

In this regard, how do the authors explain that high HOXB T-ALL cases, like N-only leukemias, display a late thymic progenitor phenotype, while NLTB leukemias with elevated HOXB activation are similar to early progenitors?

>> This is an interesting point for which no obvious or simple explanations present themselves. We might offer the observation, however, that while HOXB high cases are relatively enriched for later stage phenotypes, they do exhibit a spectrum of phenotypes including a minority with earlier stage phenotypes. Notable in this regard are two HOXB high cases that classify as ETP-ALL (**Fig 7a,b**). Similarly, NLTB leukemias also exhibit a spectrum of phenotypes including DN, DP, and dim SP8 (**Fig S3** and **Table S1**). Thus, we can only surmise that HOXB expression tends to be associated, but is not inexorably linked to the overall stage of differentiation in this context. Importantly, we have revised the results text describing immunophenotyping/TCR rearrangement studies of NLTB leukemias to emphasize the spectrum of pre- and post- β -selection phenotypes observed.

5) In Figure 7c,d,e, proportions of shScr-transduced control cells decrease along culture, suggesting a non-specific impact of the shRNA transduction procedure on in vitro cell proliferation. This is particularly important regarding Fig. 7e data, which may support the claim that HOXB genes are crucial for the initiation of leukemia. Thus, formal proof should be provided that freshly transduced CB control cells in Fig. 7e do actually generate leukemia in contrast to shHOXB3-transduced cells.

>> We agree on this point that both shScr and shLuc controls do indeed have measurable negative impacts on cell growth in transduced cells. Despite this effect, the data clearly show that multiple shRNAs against HOXB3 negatively impact cell growth to a significantly greater extent in every context tested. Nonetheless, we agree that data showing effects in freshly transduced CB cells (as in the original **Fig 7e**) is critical to our conclusion on the role of HOXB3 in leukemia initiation. While we appreciate that injecting shScr vs shHOXB3-transduced CB cells (in combination with N+LTB viruses) into recipient mice would be the most direct way of confirming that shScr cells are indeed capable of generating leukemia, this experiment would take approximately 6-8 months to complete (~6 months latency for NLTB/shScr leukemias to

develop, then an additional ~2 months to show that mice injected with NLTB/shHOXB3 cells do not develop leukemia).

To get at an answer as to the effect of shHOXB3 on leukemia initiation in a more expedient fashion, we performed an *in vitro* test of clonogenic activity that is very similar to the conventional colony forming cell (CFC) assay, but is adapted to be compatible with OP9-DL1 co-cultures. Briefly, freshly transduced CB cells are sorted into individual wells of a 96-well plate that has been pre-plated with OP9-DL1 feeders at limiting dilution (doses of 100, 50, 25, 10, 3, and 1 cell per well). After ~3 weeks of culture, each well is assayed by flow cytometry (with counting beads) to determine net cell yield and to confirm continued expression of the transduced markers. This assay, which we have termed the “well initiating cell”, or WIC assay, has the added benefit over transplantation of bulk cell populations in that it is more informative with respect to clonal variation among the transduced target cell population, and in this regard, gives a more robust measure as to the proliferative capacity of individual transduced cells. As shown in the new **Fig 8e/f**, transduction with two different HOXB3 shRNAs results in a statistically significant, 6- to 12-fold decrease in WIC activity as compared to shScr-transduced controls. We express these data both in terms of cell yield per well (**Fig 8e**) and the number of wells yielding at least 500 triply-transduced (N/GFP+, LTB/Cherry+, shRNA/NGFR+), hCD45+ progeny (**Fig 8f**). Similarly significant results were obtained using alternate cell yield cutoffs down to 200 cells/well (data not shown). We feel these new data represent compelling evidence supporting the notion that HOXB3 is an important contributor to clonogenic expansion of freshly transduced CB cells, and which we feel can be reasonably taken as a requisite property for leukemia initiation.

Minor points:

In Figure S3, both G+C+ and G+C- leukemias with a DP TCRalpha phenotype are shown. How do the authors explain that most of these cells are negative for CD3?

>> Thank you for pointing this out. Indeed, TCRab+ cells should reasonably be CD3+. We have noted variable lot-to-lot performance from our anti-CD3 reagent. We thus repeated these assays and found that indeed the TCRab+ subsets are also CD3+. We have now replaced the relevant figure panels in **Fig S3**.

Why were transplanted mice boosted with IL-7/IL-7 mAb? What is the impact of IL-7 on leukemia generation/progression?

>> Our original protocol included boosting transplanted mice with IL-7/IL-7 mAb as per published work from Zuniga-Pflucker's lab showing engraftment of NSG mice by human CB-

derived T progenitor cells (Awong et al, Blood 2009; <https://doi.org/10.1182/blood-2008-10-187013>). We included the IL-7/IL-7 mAb injections for the first month post-transplant of primary recipients, but not thereafter, and never for serial transplants. It is not clear to us at this time if the IL-7/IL-7 mAb boosts are necessary for initial engraftment/leukemia establishment in NSG recipients. We hope to explore this issue in a controlled set of experiments in the future. Of note, there is reasonable *in vitro* evidence for cross-reactivity of mouse IL-7 on human cells (<https://resources.rndsystems.com/pdfs/datasheets/407-ml.pdf> and <https://www.peprotech.com/en/recombinant-murine-il-7>).

Reviewer #2 (Remarks to the Author):

Kusakabe, et al describe the generation of synthetic T-cell acute lymphoblastic leukemia by lentiviral transduction of human hematopoietic cells. The authors show that the leukemias that develop phenotypically recapitulate naturally occurring T cell ALL. They go on to use their synthetic model to identify aberrant anterior HOXB gene expression in their model and correlate with primary T ALL sample data. The model system described here is of interest, as creating genetic models of leukemia in human cells has obvious advantages over use of other model systems such as murine models, PDX models and human cell lines. In general the paper is well written and the figures are of high quality. However, a few issues should be addressed before the paper is suitable for publication.

1. The authors only present data from N+LTB transduction, but other combinations of were also performed. Do these other synthetic leukemias similarly relatively overexpress anterior HOXB cluster genes? If so rather than be specific to LTB transduction, perhaps this pattern of expression is just inherent to synthetic T cell leukemia derived from cord blood stem/progenitor cells.

>> This is an interesting question we had not previously considered. Analysis of additional new RNA-seq data reveals that indeed upregulation of multiple HOXB genes does appear to be unique to LTB as this feature is not observed for N in combination with any of the other oncogenes (LYL1, TLX1, TLX3, HOXA9, MEF2C, NKX2.1) (Fig S9). We have included additional results text to include this new data.

The authors report that HOXB overexpressing T ALL patient samples were associated with TAL1, NKX2-1 and “unknown”. But if the authors look instead at TAL1/LMO2 cases, is there consistent overexpression of anterior HOXB genes?

>> We can answer this question in two ways, depending on how one defines “TAL1/LMO2” cases. If we consider TF subgroups as defined by Mullighan’s team (Liu et al, 2017), then TAL1, NKX2-1, and “unknown” subgroups are consistently high expressers of anterior HOXB genes, while TLX1, TLX3, and HOXA are consistently low expressers (Fig S11). The remaining subgroups LMO1/2, LMO2/LYL1, and TAL2 are low for HOXB2 and intermediate for HOXB3/4. There is very little, if any expression seen for HOXB5 in any of the subgroups.

If we instead examine cases in the highest quintile for expression of TAL1 and LMO2 genes, respectively, we find that indeed HOXB genes are generally more highly expressed in the top quintile of TAL1 expressers, although this difference is statistically significant only for HOXB4 (Fig S12). There was no evidence for increased HOXB gene expression in the top quintile of

LMO2 expressers. Additional studies will be needed to assess what genetic contexts are most permissive to HOXB gene upregulation. We have revised the text to incorporate these additional data.

2. The exact methodologies employed by the authors in various experiments is confusing to follow. They state that for early protocols cells were harvested from OP9 culture at 10 days and transplanted intrahepatically into neonatal mice, but later experiments used sorted G+C+ cells harvested at 24-25 days. In figure 2, presumably mice from strategy 1 are depicted, but this is not made clear. Why were the methods changed? Presumably most of the data presented is from strategy 1 given leukemias of G+C+, G+C- and mixed populations were documented from the same sample as shown in figure 2a, and gene expression from these differing types of leukemia were used to generate gene expression analyses comparing G+C+ to G+C- leukemias.

>> We apologize for any confusion as to the two approaches. Indeed **Fig 2a** contains data for both approaches. Full details for every primary transplant recipient are included in the associated **Table S1**. We left out the d10 vs d24 notation from the legend of **Fig 2a** for clarity, but have now included it to address this ambiguity. As can also be seen from **Table S1**, both G+C+ and G+C- leukemias were obtained from d10 injected samples, while sorted G+C+ cells from d24 cultures most invariably yielded G+C+ leukemias.

In our first set of attempts to generate CB leukemias, we only obtained enough cells by d10 of culture to inject a few recipient mice due to low percentages of doubly transduced (G+C+) cells. We therefore continued a fraction of the d10 cultures until d24, at which time the G+C+ fraction had increased by over 10-fold, thus allowing greater numbers of recipient animals to be injected.

3. In figure 1e, the legend states that the plotted data are from either 19 or 33 days in culture post transduction. Please indicate which points are from 19 and which from 33. Are these from the same samples, analyzed at two time points? If so, is there any change in immunophenotype over time?

>> We have now split out the data into separate columns for d19 and d33 samples (**Fig 1e**). Three of the 4 datapoints for d33 also have data for d19. As can be seen now that the data for different days is segregated, the percentage of CD34+ CD38+ cells decreases over time for both G+C+ and G-C- subsets. There is no change in the percentage of CD7+ CD1a+ cells over time for the G+C+ subset, but variable increase for the G-C- subset.

4. In figure 3a, the legends states that there were STR patterns from 4 different individual

donors, whereas in the text it states in line 158 that ‘at least 5 different STR profiles’ were present. Please clarify.

>> Thank you for pointing this out. The 5th donor was originally referred to as “data not shown”, but we have now included data for the 5th donor as **Fig S6** and revised the text to clarify this point.

5. In figure 4, it would be helpful to show the corresponding RNAseq tracks.

>> We have now included the RNAseq tracks as suggested (**Fig S10**).

6. In figure 5C and 5D, it appears there were 51 patients with high HOXB3 expression and 51 with high HOXB4 expression. Is this just coincidentally the same number of patients (some presumably the same, but some must differ because the curves are different)? Maybe this is the correct number for both, but just want to verify.

>> The 51 patients with high HOXB3 are not identical to the 51 with high HOXB4 (34 are common to both, 17 are unique to one or the other). The reason there are always 51 “high” cases out of 252 patients for each gene is that “high” is defined as above the 80th percentile rank for each respective gene. **Table S9** lists all patients in the high “B2 or B3 or B4 or B5” group, with highlighted entries indicating which individual HOXB genes were above their respective 80th percentile cutoffs.

7. In supplemental figure S3B, please use an arrow to indicate the bottom row is gated G+C-.

>> Thank you for catching this inadvertent omission. The appropriate arrow has now been added.

8. In line 99, self-cleaving should technically be in quotations, since the process is ribosome skipping, not actual self-cleavage.

>> We have inserted quotation marks as suggested.

9. Suggest avoiding the use of the phrase “just missed the $p < 0.05$ cutoff” in line 246. Rather just state it was not statistically significant.

>> We have edited the text as suggested.

10. Rather than “packed” bone marrow in line 128, I suggest the authors use a more objective description of the bone marrow morphology such as hypercellular or infiltrated with leukemic

blasts, for example.

>> We have edited the text as suggested.

Reviewer #3 (Remarks to the Author):

In this manuscript, Weng et al report a synthetic model of T-cell leukemia and discover the role of HOXB genes in its initiation and maintenance. More specifically, to show the functional importance of these factors, the authors perform a focused shRNA drop-out pooled screen in NLTB leukemia and T-ALL cell lines. The reviewer has several questions and/or suggestions regarding this part of the manuscript:

1. The methods section describing the screen is written with insufficient detail. Has the custom-made pooled library been sequenced to check for initial representation of the shRNA? If not, how did the T0 representation look like?

>> We apologize for this oversight. We have now provided additional methodological details for the screen including details on the library construction and verification. We did indeed sequence the custom-made pooled plasmid library and found all input clones could be detected with ~55% of clones within 4-fold and ~75% within 10-fold of the mean read counts. All input clones were also detected at t0 in both trials of the primary CB leukemia screen with ~55% of clones within 4-fold and ~90% of clones within 10-fold of the mean read counts (**Fig S14**).

How were the transduced cells selected for?

>> As described in the revised methods section, we did not apply selection for the shRNA puro marker in order to minimize the time duration between initial shRNA transduction and collection of the t0 time point sample. We elected instead to maintain larger numbers of cells in culture (including both transduced and non-transduced populations) and process the larger mass of genomic DNA over multiple separate PCR reactions.

Please state how many biological replicates were performed. What was the correlation between independent replicates

>> We performed two independent replicates of the 59-plex shHOX screen. The Spearman correlation coefficient between replicates was 0.58. We have added this information to the legend in **Fig 8b**.

2. The authors state in the main text that they conducted the same screen in T-ALL cell lines with HOXB3 showing consistent drop out. Please show volcano plots in a supplementary file.

>> We performed the same 59-plex shRNA screen on two different cell lines, HSB2 and PEER, each in single replicate. This design does not allow for generation of volcano plots due to

variation between the two cell lines, but we have now provided fold-change data for each of the shRNA species in **Fig S16** which shows consistent depletion of HOXB3 shRNAs 643, 644, and 647, confirming results from the primary leukemia screen. Moreover, the individual shRNA tracking experiments (now repeated in triplicate assay) confirm consistent selection against shHOXB3.

3. In the primary screen, why was T1 collection point chosen to be at Days 9-11? Typically longer time intervals are needed for negative selection screens to achieve a better dynamic range.

>> We elected to run the primary HOX shRNA screen for only 9-11 days based on preliminary testing that showed this was sufficient to yield 5 population doublings. It is useful to keep in mind that these are primary leukemia cells and not adapted to long-term culture *in vitro*. As such, net culture yields tend to flatten out over time, and thus running the assay for longer periods yields diminishing returns and increased risk of spurious clone loss.

Are these cells proliferating much faster compared to T-ALL cell lines where it took longer for shHOXB3-transduced cells to drop out (e.g. Figure 7D)?

>> We would estimate the doubling time for 105_LEP cells on feeders to be ~43 hours, which is comparable to that for many established human T-ALL cell lines which range from 25-60 hours (DSMZ website). The apparent rapidity of shRNA dropout for primary leukemia cultures (eg original **Fig 7c**) as compared to cell lines (eg original **Fig 7d**) was not reproduced in subsequent replicate assays (see revised **Fig 8c/d**).

4. It appears that validation experiments in figure 7C are performed in another NLTB cell line compared to the one used in the primary screen. If this is correct, it would be important to show validation using the cell line from the screen. These screens are inherently noisy and validation is critical.

>> We have now performed the individual shRNA/GFP tracking experiment with the same primary leukemia as was used for the 59-plex shRNA screen. These new data are included in the revised **Fig 8c**

Are the 3 shRNAs against HOXB3 scored as hits in the screen the same as the ones outlined in figure 7C, D and E?

>> Yes they are the same 3 shRNA clones. We have now annotated **Fig 8b** more clearly to include this information.

5. As part of the validation protocol, it is also important to show that shRNA act on-target. A western blot and/or qPCR showing an extent of knock down using the constructs against HOXB3 should be shown in a supplementary file. It would be also interesting to check the performance of HOXB5 shRNA to eliminate potential lack of efficacy of these shRNAs.

>> We had included western blot data showing knock-down efficiency for the shHOXB3 clones in the original **Fig S6**. We now include western blot data showing knock-down efficiency for both HOXB3 and HOXB5 shRNA clones in the new **Fig S15**.

6. Graphs in figures 7C, D, and E do not show any error bars. Please display error bars at least for technical replicates or if they were not performed state why not.

>> We have now repeated the experiments from original **Figs 7c/d** in triplicate and replaced the original figure panels with these new data (revised **Figs 8c/d**). Error bars showing SD are included as well.

In response to a comment from Reviewer #1 regarding the HOXB3 knock-down experiment with freshly transduced CB cells (original **Fig 7e**), we elected to perform a limiting dilution growth assay instead of bulk culture in order to track the growth of individually transduced CB progenitors (similar to a conventional colony forming cell, or CFC assay). We have thus replaced the original data from **Fig 7e** with the limit dilution experiment in the new **Fig 8e/f**.

REVIEWERS' COMMENTS:

Reviewer #1 (Remarks to the Author):

The authors have appropriately addressed all raised concerns

Reviewer #2 (Remarks to the Author):

The authors have adequately addressed my comments and the comments of other reviewers. I have no further comments.

Thank you for this interesting report.

Sincerely,

Rachel Rau

Reviewer #3 (Remarks to the Author):

This reviewer is satisfied with how the authors addressed all of the raised comments and can recommend the manuscript for publication pending other reviewers' assessment.

This reviewer would also like to thank the authors for making an effort to include all the relevant technical details with respect to the screens performed, which is necessary for upholding a transparent and reproducible science.

Iva Nikolic